# InkSight: Offline-to-Online Handwriting Conversion by Teaching Vision-Language Models to Read and Write

◇**Blagoj Mitrevski**[Ⓔ], ♡**Arina Rak**[Ⓔ], ♡**Julian Schnitzler**[Ⓔ], ♡**Chengkun Li**[Ⓔ], ◇**Andrii Maksai**[✉],
◇**Jesse Berent**, ◇**Claudiu Musat**

◇Google DeepMind, ♡EPFL, work done as student researcher.
[Ⓔ]First authors: random order decided by AEA tool, [✉]Project lead: amaksai@google.com

Reviewed on OpenReview: https://openreview.net/forum?id=pSyUfV5BqA

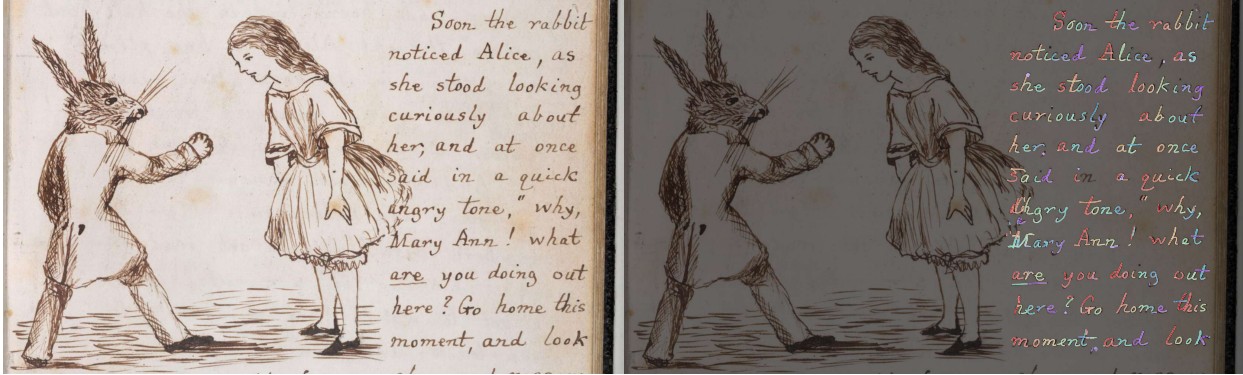

Figure 1: **Results of InkSight** [Animated visualization]. **Left:** A photo of handwritten text (offline handwriting), **Right:** Output digital ink (online handwriting). In every word, character colors transition from red to purple, following the rainbow sequence, ROYGBIV, which reflects the stroke order; within each stroke, the shade progresses from darker to lighter. More visualizations in Appendix B.

## Abstract

Digital note-taking is gaining popularity, offering a durable, editable, and easily indexable way of storing notes in a vectorized form, known as digital ink. However, a substantial gap remains between this way of note-taking and traditional pen-and-paper note-taking, a practice that is still favored by a vast majority. Our work InkSight[1], aims to bridge the gap by empowering physical note-takers to effortlessly convert their work (offline handwriting) to digital ink (online handwriting), a process we refer to as **derendering**. Prior research on the topic has focused on the geometric properties of images, resulting in limited generalization beyond their training domains. Our approach leverages vision-language models to combine *reading* and *writing* priors, allowing training a model in the absence of large amounts of paired samples, which are difficult to obtain. To our knowledge, this is the first work that effectively derenders handwritten text in arbitrary photos with diverse visual characteristics and backgrounds. Furthermore, it generalizes beyond its training domain into simple sketches. Our human evaluation reveals that 87% of the samples produced by our model on the challenging HierText dataset are considered as a valid tracing of the input image and 67% look like a pen trajectory traced by a human.

## 1 Introduction

Handwritten notes have been a cornerstone of information storage for centuries. Today, with the rise of stylus and digital pen technologies, digital inking is a compelling alternative. This modern approach to

---

[1]GitHub: https://github.com/google-research/inksight Hugging Face: [link] Project page: [link]

storing information offers several advantages: Digital ink offers enhanced durability, structured organization, and fine-grained editability. Unlike pen-and-paper notes, it supports post hoc modifications users can move, resize, or restyle individual strokes or sections. This enables dynamic reuse, seamless integration with digital workflows, and compatibility with assistive systems. Despite these apparent benefits, many people still prefer traditional handwritten notes over the digital format.

Traditional OCR extracts textual content from handwriting but discards the rich structural and temporal information embedded in the handwritten form. In contrast, our approach captures handwriting as digital ink, preserving stroke-level trajectories. This enables users to retain the familiarity of pen-and-paper note-taking while gaining the flexibility of digital representations without requiring a stylus.

The topic of converting offline handwriting to online handwriting has gained significant interest in both academia (Nguyen et al., 2021; Chen et al., 2022b; Mohamed Moussa et al., 2023) and industry (Not, 2023), with the introduction of software solutions that digitize handwriting and hardware solutions that require smart pens and/or special gadgets (Nuw, 2025; Liv, 2024; Roc, 2024; Neo, 2024). Our approach requires only an image of the handwritten note, without any specialized equipment. It differs from the methods that rely on geometric priors, where gradients, contours, and shapes in an image are utilized to extract writing strokes. Instead, we harness the power of learned *reading* and *writing* priors, where:

- The learned *reading* prior enhances the model's capability to precisely locate and extract textual elements from images. This is achieved either through the model's textual understanding ability or aided with external textual input, e.g. from an OCR engine.
- The integration of the *writing* prior ensures that the resulting vector representation, the digital ink, closely aligns with the typical human approach to writing in terms of physical dynamics and the order of strokes.

To the best of our knowledge, our work is the first to incorporate these priors, resulting in a robust model that is capable of derendering handwriting across diverse scenarios and appearances, including challenging lighting conditions, noticeable occlusions, etc. Our model adopts a simple architecture combining the widely popular and readily available ViT (Dosovitskiy et al., 2021) encoder and an mT5 (Xue et al., 2021) encoder-decoder, fostering reproducibility, reusability, and ease of adoption. To summarize, the major contributions of our work are:

1. We propose the first system to perform **derendering**: transforming arbitrary photos of handwritten text into digital ink. It relies only on the standard architecture components and the training and inference setup that works without expensive data collections and scales to arbitrarily large input images.
2. We show that the inks produced by our system are both semantically and geometrically similar to the input images. We demonstrate that they are similar to real digital ink data, as measured by both automatic and human evaluations.
3. We show that, with the learned reading and writing priors, our approach is robust and works on various types of handwriting, simple sketches, and full pages of notes.
4. We release the public model Small-p, trained entirely on publicly available datasets, along with a subset of model-generated inks in universal `inkML` format and expert-traced online handwriting data to support future research on the topic.

## 2  Related Work

**Pen trajectory recovery** has been a task of interest to many researchers due to its utility for tasks that can benefit from stroke order / temporal information such as online handwriting recognition (Lallican et al., 2004; Viard-Gaudin et al., 2005; Zhang et al., 2015; Chan, 2020). Another point of interest is the ability to efficiently store, index, and edit handwritten notes within existing online note-taking systems. Classical approaches commonly consist of domain-specific preprocessing (e.g. noise removal, image binarization, skeletonization), local (sub-)stroke level processing (e.g. identification of junction points) and global aggregation (commonly as a graph-based optimization problem) (Jager, 1996; Kato & Yasuhara, 2000; Qiao et al., 2006; Chan, 2020; Doermann et al., 2002). These approaches often rely on the quality of the preprocessing and hand-

designed heuristics and do not generalize well to other scripts and domains. More recent methods use convolutional (Nguyen et al., 2021; Archibald et al., 2021) and/or recurrent (Bhunia et al., 2018; Chen et al., 2022b) neural networks to translate an image into a sequence of coordinates. Set, sort! (Mohamed Moussa et al., 2023) use two Transformer models to first encode the substrokes of the input and then reorder them, given the encodings. Those methods produce promising results, but focus on the simplified setups of rendered online handwriting and/or single characters.

**Line drawing vectorization** shares many features with geometric approaches to pen trajectory recovery, but does not include stroke order reconstruction. Recent techniques for this problem involve solving an optimization task of fitting a set of geometric primitives (e.g. Bezier curves) to match the geometric and/or semantic content of the input image (Vinker et al., 2022), sometimes relying on differentiable rendering to maintain the inputs and outputs in the image space (Mo et al., 2021). These approaches focus on converting raster images (perfectly aligned, clean, clear backgrounds) to vector ones. Thus they avoid dealing with the realistic photo artifacts related to lighting, noisy backgrounds or photo-taking angles. These artifacts are however unavoidable in a real-world setting.

**Dataset availability** is limiting the research in trajectory recovery as there are few datasets of images and their corresponding digital inks. These include IRONOFF (Viard-Gaudin et al., 1999), CASIA (Liu et al., 2011), IBM-UB (Shivram et al., 2013) for handwritten text (images there exhibit limited variability being always black-on-white line writing, usually on the same input device) and Sketchy (Sangkloy et al., 2016) for crowd-sourced sketches of photos from 125 object categories.

**Combining language models with new modalities** is related to this work since we convert the pen trajectory to a sequence of coordinates on a fixed size canvas and represent the coordinates with *ink tokens* (more details in Section 3.3). Hence, we consider our approach as extending a Visual Language Model (VLM) with a new modality. Recent research in this area has been pointed towards training adapters for merging the pretrained unimodal models (e.g. Alayrac et al. (2022)) or, fully or partially fine-tuning a system with multi-modal tasks (e.g. Chen et al. (2022a)). Works most similar to ours are AudioPaLM (Rubenstein et al., 2023), extending LLM with audio tokens, Painter (Pourreza et al., 2023) using coordinate-based representation of strokes to perform generation and understanding of sketches, and PixelLLM (Xu et al., 2024) using the same representation to localize words in the image caption.

## 3 Method

While the fundamental concept of derendering appears straightforward – training a model that generates digital ink representations from input images – the practical implementation for arbitrary input images presents two significant challenges:

1. **Limited Supervised Data:** Acquiring paired data with corresponding images and ground truth digital ink for supervised training can be expensive and time-consuming and no datasets with sufficient diversity exist for this task yet.
2. **Scalability to Large Images:** The model must effectively handle potentially arbitrary large input images with varying resolutions and amounts of content.

To address the first problem without expensive data collection, we propose a multi-task training setup that combines recognition and derendering tasks. We show that this setup enables the model to generalize on derendering tasks with various styles of images as input, and injects the model with both semantic understanding and writing priors of handwritten text. We discuss our data and multi-task training setup in Section 3.2, and ablate the design choices in Section 4.6. The exact model structure, which consists of standard publicly available off-the-shelf components, is discussed in Section 3.1. In Section 3.3, we cover the representation of digital ink, the normalization of the data, and the tokenization scheme we employ.

The second problem can be addressed by training a model with very high-resolution input images and very long output sequences, but this is computationally prohibitive. Instead, we break down the derendering of a page of notes into three steps: running OCR to extract word-level bounding boxes, derendering each of the words separately and pasting the derendered words back, as shown in Figure 2. An example of full page inference is shown in Figure 1, more full page results are presented in Appendix N.

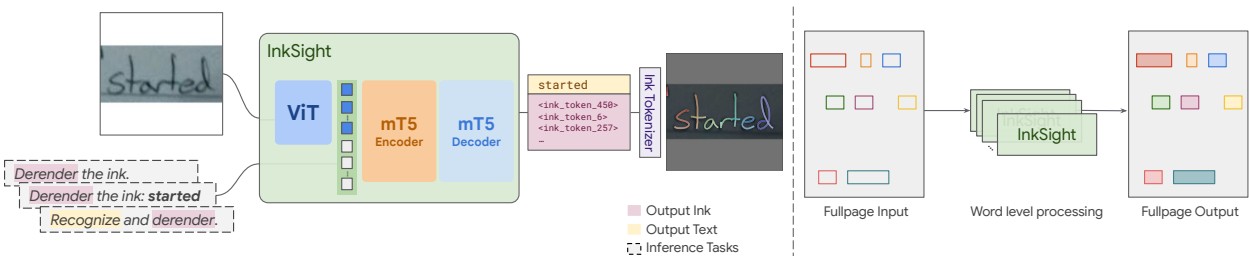

Figure 2: **Left [Animated visualization]:** Illustration of model inference. Inputs include an image and a prompt specifying the task. Outputs may consist of ink alone or a combination of ink and text, as indicated by matching colors in both the input prompt and the output. Ink tokens are detokenized into digital ink using our proposed ink tokenizer explained in Section 3.3. **Right [Animated visualization]:** Diagram of the Full Page System.

## 3.1 Vision-Language Model for Digital Ink

Our model architecture, inspired by that of PaLI (Chen et al., 2022a; 2023b;a), integrates a Vision Transformer (ViT) encoder (Dosovitskiy et al., 2021) with an mT5 encoder-decoder Transformer model (Xue et al., 2021). We provide the task-specific instructions as text, as described in Section 3.2. We use the same set of tokens for both input and output, which contains the individual character tokens from the original mT5 vocabulary and specific tokens reserved for the ink representation (details in Sec. 3.3). To initialize the model, we use the pre-trained weights of the ViT encoder. The mT5-based encoder-decoder weights are initialized randomly, which is motivated by our customized token dictionary differing from the one used in mT5's original training. In our training, we freeze the weights of the ViT encoder – we justify this choice empirically in Section 4.6.

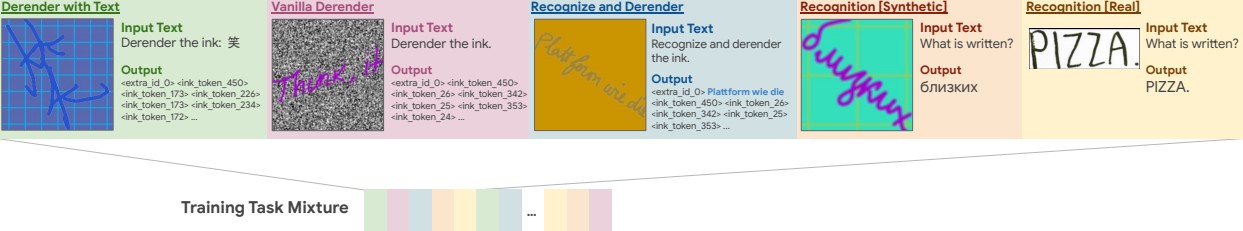

Figure 3: **Illustration of multi-task training mixture.** The training mixture comprises five different task types: two derendering tasks (ink output), two recognition tasks (text output), and one mixed task (text-and-ink output). Each type of task utilizes a task-specific input text, enabling the model to distinguish between tasks during both training and inference.

## 3.2 Training Task Mixture

To circumvent the challenge of not having diverse paired image and ink samples as training data, we propose a multi-task training setup comprising two derendering tasks, two recognition tasks, and one hybrid task (shown in Figure 3 and described in Table 1).

We show that this setup helps the model to: (1) generalize to the derendering of real photos; (2) learn useful priors that help deal with occlusions and generate realistic ink; (3) allow using different inference setups that enable, e.g. high-quality text derendering with OCR result (*Derender with Text*) or derendering without textual input for non-textual elements (*Vanilla Derender*). Training tasks are shuffled and assigned equal appearance probability (20%). However, future work could explore adjusting this ratio dynamically during training and towards the end of training. For best performance during inference, we use *Derender with Text* (this choice is ablated in Section 4.6 where we show that our model performs well without reliance on OCR as well).

Table 1: Summary of tasks in our proposed multi-task training.

| Task | Description |
| --- | --- |
| Vanilla Derender | The model receives a synthetic ink image with a conditioning task prompt and outputs ink tokens corresponding to the image content. |
| Derender with Text | The model receives a synthetic ink image and a task conditioning prompt including the target textual content, and outputs ink tokens corresponding to the provided text within the image. |
| Recognition (Syn/Real) | The model receives a handwriting image (either synthetic or real) and a task conditioning prompt and outputs the recognized textual content. This task uses both synthetic ink images and real images from OCR datasets. |
| Recognize and Derender | This task combines recognition and derendering. The model receives a synthetic ink image and a prompt and outputs both the recognized text and the corresponding ink tokens. |

**Data augmentation.** To narrow the domain gap between the synthetic images of rendered inks and the real photos, we augment the data in tasks that take rendered ink as input. This is done by randomizing the ink angle, color, stroke width, and by adding Gaussian noise and cluttered backgrounds. Examples are provided in Figure 3, more details are given in Appendix F, and Section 4.6 highlights the importance of having the data augmentation.

### 3.3 Data representation

**Digital Ink tokenization.** Digital ink is usually represented as a sequence of strokes $I = \{s_1, s_2, \cdots, s_n\}$, where each stroke $s_i$ consists of a sequence of $m_i$ (length of the $i$-th stroke) coordinate-time triplets, represented as $s_i = \{(x_i, y_i, t_i)\}_{i=1}^{m_i}$. To enable sequence models like mT5 to generate ink strokes, we first normalize the ink and then tokenize it into a set of discrete tokens from a pre-defined dictionary. Normalizing the ink includes:

1. **Resampling:** Resample the ink at a fixed frequency (20 ms) to account for potential variations between input devices.
2. **Sequence Reduction:** Apply Ramer-Douglas-Peucker resampling (Visvalingam & Whyatt, 1990) to reduce the sequence length while preserving the overall shape of the strokes.
3. **Normalization:** Shift and scale the ink strokes to position them at the center of a fixed-size canvas. Each point in a stroke is scaled to fit within the range $[0, N]$, where we use $N = 224$ in our implementation.

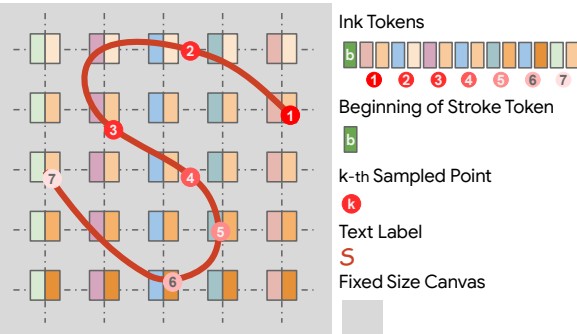

Figure 4: **Ink tokenization for a single-stroke digital ink [Animated visualization].** The dark red ink depicts the normalized ink stroke, with numbered circles marking sampled points after time resampling. The color gradient of the sampled points indicates the point order. Each point is represented with two tokens encoding its $x$ and $y$ coordinates. The token sequence for this ink begins with b, signifying the beginning of the stroke, followed by the tokens for coordinates of sampled points.

As shown in Figure 4, tokenization of the normalized ink involves prepending a ***beginning of stroke*** token followed by the $x$ and $y$ coordinates of each point inside the stroke, rounded to the nearest integer value.

This is done for every stroke inside the digital ink. The total dictionary size is $2N + 3$ with separate set of tokens for $x$ and $y$, including $N + 1$ possible $x$ and $y$ values and a token indicating a start of the stroke. $N$ controls the tradeoff between rounding error and vocabulary size, and we use $N = 224$ in practice.

**Image representation.** For the derendering tasks of the training mixture, we render digital inks in the center of an $M \times M$ image with stroke width, color, and background color selected via random augmentation (see Appendix F). For the recognition task training and inference with real images, we apply a similar transformation: the input image is scaled, centered, and padded in black to produce an $M \times M$ image. Although we use $M = 224$, by design there are no constraints to have $N = M$.

**Expanding text vocabulary with Ink.** Our set of tokens contains all individual symbols from the multilingual mT5 tokenizer (Xue et al., 2021) (approx. 20k), with additional $2N + 3$ tokens reserved for the representation of the digital ink, as described above. Removing all multi-symbol tokens allows to reduce the size of the input text embedding and final softmax of the decoder model by $\sim 80\%$, while maintaining the ability to input and output arbitrary text and not affecting the quality of derendering or recognition.

## 4 Experiments

In this section, we discuss the training datasets and implementation details, and present the qualitative and quantitative results, followed by an ablation study on training tasks and design choices.

### 4.1 Datasets

We train two different models using two types of datasets: one trained only on publicly available data and one trained on our in-house proprietary data. Below, we detail the public datasets and provide aggregated statistics for the in-house counterparts. Additional details are provided in Appendix C.

**OCR training data.** As public OCR training data, we use RIMES (Augustin et al., 2006; Grosicki et al., 2009), HierText (Long et al., 2022; 2023), IMGUR5K (Krishnan et al., 2023), ICDAR'15 historical documents (Murdock et al., 2015), and IAM (Marti & Bunke, 1999). We crop out word-level images where possible, acquiring 295 000 samples with text in Latin script. The in-house dataset consists of images of handwritten and printed text (67% and 33% respectively) written on a diverse set of backgrounds, with 95% of the labels in English.

For data sources that contain word-level segmentation, we extract images of individual words based on their bounding boxes. For others, we found it beneficial to heuristically filter the training data to exclude samples that are too short, too long, or of too low resolution to be rendered on a $224 \times 224$ image. The filtering rules ensure that the aspect ratio satisfies (0.5 < width / height < 4.0), and that the image size is at least 25 pixels per side.

**Digital ink training data.** As public digital ink training data, we use VNOnDB (Nguyen et al., 2018), SCUT-Couch (Li et al., 2008), and DeepWriting (Aksan et al., 2018). For DeepWriting we use the available segmentation information to produce 3 datasets with character-level, word-level, and line-level croppings, and use all 3 for training. For VNOnDB we extract individual words to be used as training data. The public dataset size totals $\sim$2.7M samples.The in-house multilingual dataset primarily comprises short snippets of text in Mandarin (37%) and Japanese (23%), while the remaining languages each contribute less than 5%.

**Evaluation Data.** Since there are no available diverse datasets of paired inks and images (as discussed in Section 2), we perform the evaluation on OCR data and additionally do a small data collection to obtain paired samples. To automatically assess our models, we evaluate the quality of derendering on the test splits of 3 OCR datasets: IAM (`testset_f`, $\sim$17.6k samples), IMGUR5K ($\sim$23.7k samples), and HierText. For HierText, which was not originally designed around handwritten samples, we apply the same filtering as to the OCR training data, and additionally only consider words marked as handwritten ($\sim$1.3k samples).

Additionally, we have performed a small annotation campaign, asking people to trace ∼200 samples from the HierText test set. We use these as the "golden" set to find the reference point in the automated evaluation and as a control group when doing the human evaluation.

## 4.2 Models

We train 3 variants of our model: **Small-p** (∼340M parameters, **-p** stands for **p**ublic setup) trained on publically available datasets with a ViT B/16 encoder pretrained on ImageNet-21k (Deng et al., 2009) paired with an mT5-base encoder-decoder; **Small-i** (**-i** stands for **i**n-house setup), built with the same architecture as Small-p, but using a JFT-300M (Sun et al., 2017) pretrained ViT B/16 checkpoint; **Large-i** (∼1B parameters) with a ViT L/16 encoder pretrained on JFT-300M, paired with an mT5-large encoder-decoder. All models employ a context length of 1024 for the output, and 128 for the input. Implementation details are provided in Appendix G, and a model card for Small-p in Appendix M.

## 4.3 Baseline Method

Despite pen trajectory recovery being a popular research direction in the last decades, available comparisons are limited due to the scarcity of open-sourced code, training data, and pre-trained model weights. Appendix D offers a detailed analysis of recent work and their reproducibility. We choose the General Virtual Sketching (GVS) Framework (Mo et al., 2021) as our baseline. Originally trained for sketch and photo vectorization, GVS offers a valuable reference point despite the domain discrepancy between sketches and handwritten notes. We found that providing the GVS line drawing vectorization model with binarized inputs yields best performance, we compare all 3 available models on original and binarized inputs in Appendix H.

## 4.4 Qualitative Evaluation

We compare the performance of our models and GVS on 3 public evaluation datasets (mentioned in Section 4.1) in Figures 5 and 6. Our models mostly produce results that accurately reflect the text content, disregarding semantically irrelevant background. They can also handle occlusions, highlighting the benefit of the learned *reading* prior, in contrast to GVS, which produces multiple duplicate strokes, and does not distinguish between background and foreground (which is reasonable, given it was trained for sketch vectorization). **Large-i** is able to retain more details and accommodate more diverse image styles.

Figure 5: Comparison between performance of GVS, **Small-i**, **Small-p**, and **Large-i** on IAM (top) and IMGUR5K (bottom), more visualizations in Appendix N.1.

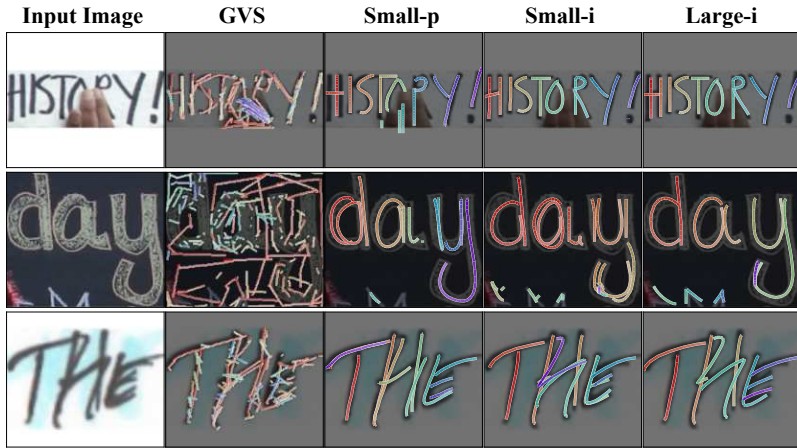

Figure 6: Comparison between performance of GVS, **Small-i**, **Small-p**, and **Large-i** on HierText, which includes data with varying styles and occlusions.

**Generalization to Sketches.** We additionally study the performance of our models on out-of-domain samples, such as simple sketches. For this, we use the *Vanilla Derender* inference mode to obtain the inks. In Figure 7, we observe that our models partly generalize to sketches, but the performance varies significantly depending on the sample and has noticeable artifacts such as missing details or over-focusing on a segment and over-tracing it. They, however, demonstrate robustness to complicated backgrounds (e.g. the lighting of the coffee sample).

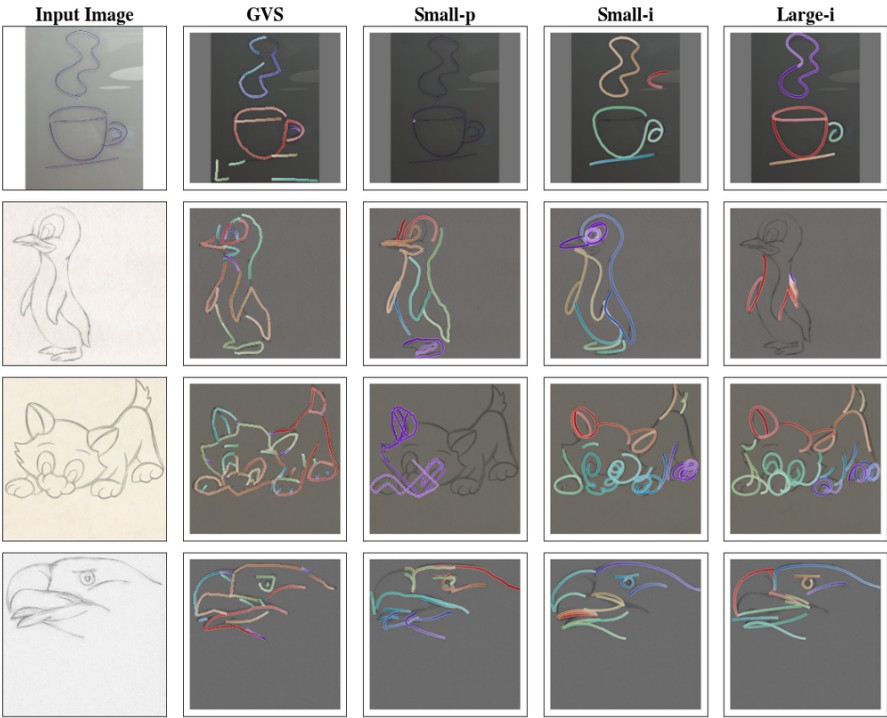

Figure 7: Sketch derendering for **Small-p**, **Small-i**, **Large-i** and GVS. Our models are mostly able to derender simple sketches, however with significant artifacts: missing strokes (e.g the cat), creating cycles (e.g the eye of the penguin for Small-i, the paws of the cat for Large-i) or unnatural strokes (e.g. for Small-p).

### 4.5 Quantitative Comparison

To support the claims about the performance of our model, we conduct both human evaluation and automated evaluation that compare the similarity of our model output to the original image and to real digital inks. To the best of our knowledge, there are no established metrics and benchmarks for this problem yet.

#### 4.5.1 Human evaluation

We performed a human evaluation of the quality of the derendered inks produced by the three variants of our model. We used the "golden" human traced data on the HierText dataset as the control group and the output of our model on these samples as the experiment group. Evaluators were shown the original image alongside a rendered digital ink, which was either model-generated or human-traced (unknown to the evaluators). They were asked to answer two questions:

1. Is the digital ink output a reasonable tracing of the input image? (Answers: Yes, it's a good tracing; It's an okay tracing, but has some small errors; It's a bad tracing, has some major artifacts);
2. Could this digital ink output have been produced by a human? (Answers: Yes; No).

We provide a more detailed description of the task and the instructions in Appendix L. The evaluation included 16 individuals familiar with digital ink, but not involved in this research. Each sample was evaluated by 3 raters and aggregated with majority voting (with inter-rater reliability $\kappa$ : (1) 0.46, (2) 0.44 for the respective questions).

We demonstrate the results of our evaluation campaign in Figure 8. We observe better performance (good or okay rating, more realistic samples) when both the data and model size are scaled up. The most common imprecisions that lead to choosing "okay tracing" over "good tracing" include: a small number of extra strokes (e. g. derendering irrelevant elements present in the image), missing details (e.g. punctuation, a dot over i, j), and unnecessary double strokes (see Appendix L for examples).

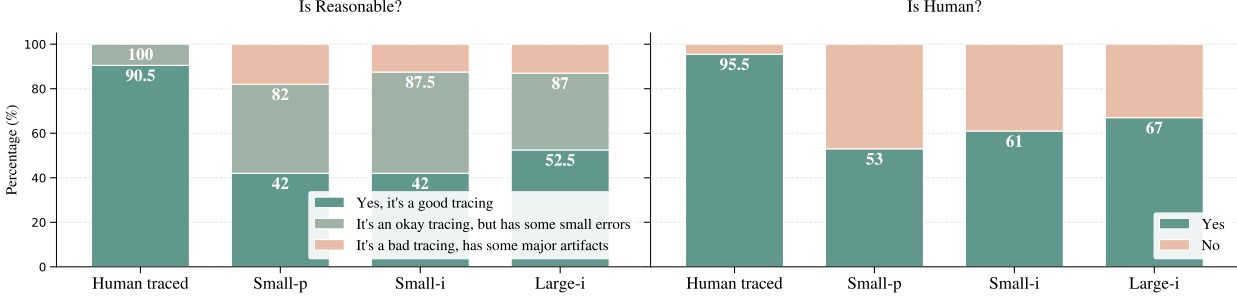

Figure 8: **Human evaluation results for Small-p, Small-i and Large-i** on 200 "golden" samples of HierText dataset.

#### 4.5.2 Automated Evaluation

Automated metrics (described below) are presented in Table 2. Most importantly, the model ranking using the automated metrics matches the results of the human evaluation.

**Similarity to the input image.** To capture how similar the output of our model is to the input image semantically and geometrically, we compute the Character Level F1 score by following the protocol of the Robust Reading Challenge (Rob, 2023). This measure captures both geometry and semantics by counting the character bounding boxes found by an OCR model (in the original input image and in the output of the model rendered as an image) that both overlap significantly and match content-wise. To obtain those character bounding boxes, we use the publicly available API (Clo, 2024). We see that our models perform similarly to GVS on simpler black-on-white IAM dataset, but outperforms it on HierText which has more diverse background. We also note that the OCR engine used is not perfect, and due to errors in the OCR model and its sensitivity to line width, the value of this metric is 0.64 for the "golden" human traced data on

Table 2: Automated metrics comparison between three variants of our model, GVS, and the "golden" human traced data. We compare both Character Level F1 score of the rendered characters and Accuracy of recognizing the generated digital ink with state-of-the-art recognizers.

| Model | IAM | | IMGUR5K | | HierText | |
|---|---|---|---|---|---|---|
| | F1 | Acc. | F1 | Acc. | F1 | Acc. |
| **Small-p** | 0.65 | 0.60 | 0.47 | 0.33 | 0.53 | 0.37 |
| **Small-i** | 0.65 | 0.59 | 0.51 | 0.33 | 0.61 | 0.44 |
| **Large-i** | 0.66 | 0.58 | 0.51 | 0.32 | 0.61 | 0.46 |
| **GVS** | 0.69 | 0.02 | 0.51 | 0.01 | 0.55 | 0.01 |
| **Human*** | – | – | – | – | 0.64 | 0.74 |

* Computed on the human-traced subset of HierText data.

HierText. Therefore, it can only be used to meaningfully distinguish models as long as their F1 score is fairly low.

**Similarity to the real digital ink data.** To assess the semantic consistency and geometric properties of our generated digital inks, we compute recognition Exact Match Accuracy with the state-of-the-art online handwriting recognizer trained on real digital ink data (Clo, 2024). As shown in Table 2, our models demonstrate superior accuracy compared to GVS (on which the online handwriting recognizer is failing completely due to unnatural stroke number, length and order). We also observed that while the three model variants perform similarly on the IAM and IMGUR5K datasets, the **Large** variant outperforms **Small** variants on the HierText dataset which has greater diversity, which aligns with our human evaluation.

To further validate the similarity between real and derendered digital inks, we train an online handwriting recognizer on inks derendered from the IAM training set. Testing on the `testset_f` of IAMOnDB (Liwicki & Bunke, 2005) and compare with the performance of model trained on IAMOnDB train set, in terms of Character Error Rate metric (CER). As shown in Figure 9, the model trained only on derendered inks achieves slightly worse CER performance compared to the model trained on real data. Furthermore, the combination of derendered IAM inks and real IAMOnDB inks allows to acquire a more diverse training set, resulting in significantly lower CER. This results holds for **Small-p** and **Large-i** as well (see Appendix I). While our CER trails state-of-the-art CER due to limited training data, our work demonstrates that offline handwriting datasets can effectively augment online recognizer training. The details of the online handwriting recognizer we train are given in Appendix I.

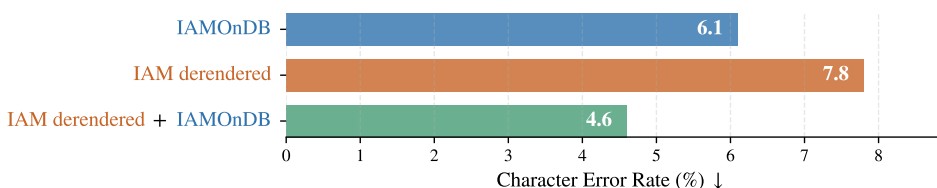

Figure 9: Online handwriting recognition results on IAMOnDB test set for Small-i trained on real digital inks (IAMOnDB train set), derendered digital inks (from IAM train set) and their combination.

## 4.6 Ablation Studies

This section examines the performance variation across inference tasks and ablates the impact of augmentation, data mixture, and training strategies on the **Small-i** model's performance. Conclusions and numerical results of the ablation study are presented in Table 3 and summarized below. For more details, see Appendix E.

Table 3: Ablation studies on **Small-i**. The first row shows the best performance setup (with inference mode *Derender with Text*). Subsequent rows display results for the *Vanilla Derender* and *Recognize and Derender* (R + D) tasks (Figure 3), followed by the impact of various design choices on performance.

| Setup | IAM | | IMGUR5K | | HierText | |
|---|---|---|---|---|---|---|
| | F1 | Acc. | F1 | Acc. | F1 | Acc. |
| **Small-i[†]** | $0.66_{\pm 0.07}$ | $0.59_{\pm 0.01}$ | $0.51_{\pm 0.09}$ | $0.33_{\pm 0.02}$ | $0.61_{\pm 0.07}$ | $0.45_{\pm 0.01}$ |
| Vanilla | 0.61↓ | 0.53↓ | 0.44↓ | 0.28↓ | 0.53↓ | 0.35↓ |
| R+D | 0.62↓ | 0.57↓ | 0.46↓ | 0.32 | 0.57↓ | 0.42↓ |
| *Remove* | | | | | | |
| data aug[*] | 0.42↓ | 0.33↓ | 0.21↓ | 0.09↓ | 0.23↓ | 0.13↓ |
| syn rec | 0.58↓ | 0.50↓ | 0.50↓ | 0.25↓ | 0.56↓ | 0.38↓ |
| real rec | 0.64↓ | 0.50↓ | 0.55↑ | 0.19↓ | 0.59↓ | 0.36↓ |
| all rec | 0.65 | 0.51↓ | 0.55↑ | 0.22↓ | 0.61 | 0.38↓ |
| frozen ViT[†] | $0.65_{\pm 0.13}$ | $0.53_{\pm 0.06}$↓ | $0.43_{\pm 0.24}$↓ | $0.31_{\pm 0.06}$ | $0.59_{\pm 0.15}$ | $0.41_{\pm 0.05}$↓ |

[*] Digital ink rendered as fixed-width white strokes on black background.
[†] The results are aggregated from three random initializations, all other results are acquired with the best performing initializations with the same three seeds.

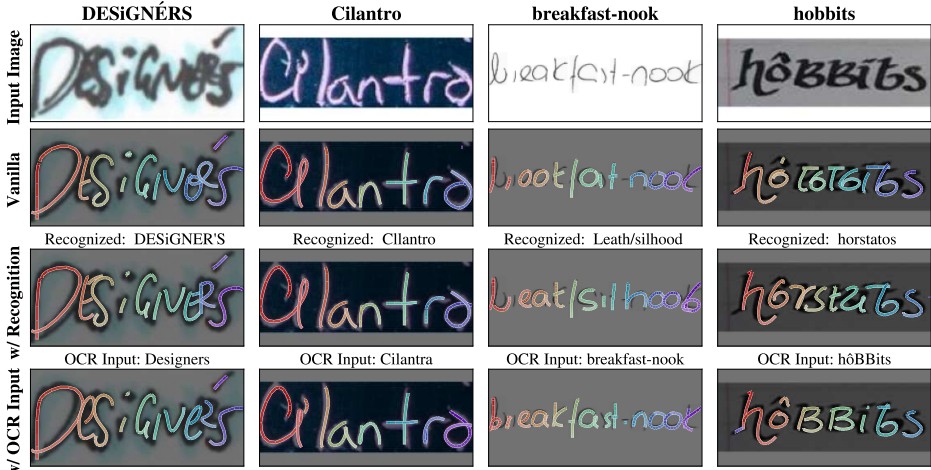

Figure 10: Varied digital ink outputs from **Small-i** inference tasks on samples with ambiguous transcription. Column titles correspond to the ground truth labels. Image titles correspond to either the recognition result from our model or an OCR system.

**Inference task matters in difficult cases (Rows 2-3).**  We compare the performance between three types of inference tasks, we find *Derender with Text* is slightly better than *Recognize and Derender* (R+D), which itself outperforms *Vanilla Derender* on all three datasets, we attribute this difference to the fact that it generates more semantically consistent. Figure 10 shows a collection of derendered inks from **Small-i** where each input image has ambiguous or difficult-to-read characters. We take the first column, DESiGNÉRS, as an example: Inference with *Vanilla Derender* captures the overall geometry of the input image but often lacks textual understanding (e.g., the letter E and ÉR are drawn as rough strokes rather than following the precise structure of the characters). *Derender with Text* produces results consistent with the text input to the model provided by OCR (upper case E and ER), while *Recognize and Derender* aligns with the model's own recognition, resulting in lowercase e and er). More examples in Appendix E.2.

**The necessity of data augmentation (Row 5).**  Removing the data augmentation leads to significantly worse performance across all metrics on all datasets. While data augmentation (as shown in Figure 3) does not visually align rendered inks with real ink photos, it diversifies the rendered ink distribution and we find that it is essential for the model to perform valid derendering on real images with our training setup.

**Recognition tasks improve derendering quality (Rows 6-8).** Removing recognition tasks from the multi-task training mixture notably reduces overall performance across all evaluation datasets, particularly impacting accuracy which reflects the model's semantic understanding ability. We hypothesize that the increase in F1 and drop in accuracy (such as the one observed on IMGUR5K) may be attributed to the model focusing less on the textual content and interpreting more background noise as strokes, which leads to digital inks that closely mimic the style of the input images, resulting in a higher likelihood of overlapping bounding boxes in F1 calculations.

**Impact of frozen ViT in multi-task training. (Last row).** Unfreezing the Vision Transformer (ViT) in multi-task training typically incurs training instability, as evidenced by increase in variance in model evaluations and an increased tendency to misinterpret background noise as textual content. Explaining the observed increase in F1 scores in Table 3 can be attributed to the model misinterpreting background noise as strokes, generating digital ink that visually closely resemble the input images, including the noise. This behavior results in a higher likelihood of overlapping bounding boxes, inflating F1 calculations. More details in Appendix E.1.

### 4.7 Limitations

We acknowledge the following two main limitations of our approach:

(1) Due to the limits in context length of transformer-based models, it can derender limited amount of characters or words in one model inference call; While our system incorporates efficient design choices such as resampling and sequence reduction to support full-page derendering, this remains a constraint.

(2) Consequently, it requires some form of segmentation of the handwritten page to run derendering on rather short individual segments (words in our implementation) and in the current form cannot work on complex sketches or other objects with lots of strokes, extending it to handle complex sketches or dense stroke-based objects presents an exciting direction for future work.

One thing that is **not** a limitation of our model is the use of OCR to provide the word labels as shown in the first 2 rows of Table 3, the model performance degrades only slightly if the label is not provided or is inferred by the model. We also discuss some specific failure cases in Appendix J.

## 5 Conclusion

In this work we present the first approach to converting photos of handwriting into digital ink. We propose a training setup that works without paired training data, which can be difficult to acquire. We show that our method is robust to a variety of inputs, can work on full handwritten notes, and generalizes to out-of-domain sketches to some extent. Furthermore, our approach does not require complex modeling and can be constructed from standard building blocks. We release the weights and inference code of the Small-p model, with a subset of model-generated and human expert-traced digital ink data. Future work may address the main limitations of our model stated above.

### Acknowledgments

The authors thank Leandro Kieliger for the help on synthetic ink rendering, feedback on initial versions of the paper, Philippe Schlattner on human evaluation protocol, Anastasiia Fadeeva, Mircea Trăichioiu for the multiple discussions, Efi Kokiopoulou, Diego Antognini, Henry Rowley, Reeve Ingle, Manuel Drazyk for feedback on initial versions of the paper, Sebastian Goodman, Jialin Wu for the help on implementing deterministic training and PaLI related questions, Xiao Wang on ViT related questions, Tom Duerig and Tomáš Ižo for leadership support.

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

# A    Appendix

## CONTENTS

*This appendix provides comprehensive supplementary materials*
*supporting the main findings and methodology presented in this work.*

# B Full-page Results

We show the full-page results on samples that resemble real-life inputs (mostly from Unsplash with keyword search for "handwriting", and others collected by the authors with consent from the writers) produced by three variants of our models.

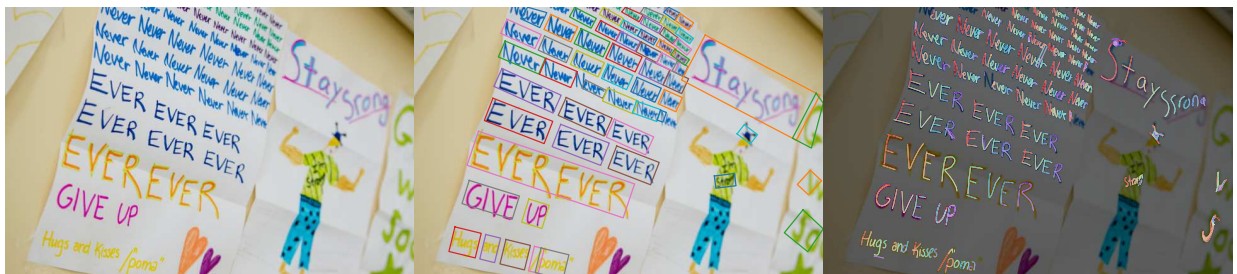

Figure 11: **Large-i** full page results of handwritten notes in a real-life scenario, credit: Unsplash.

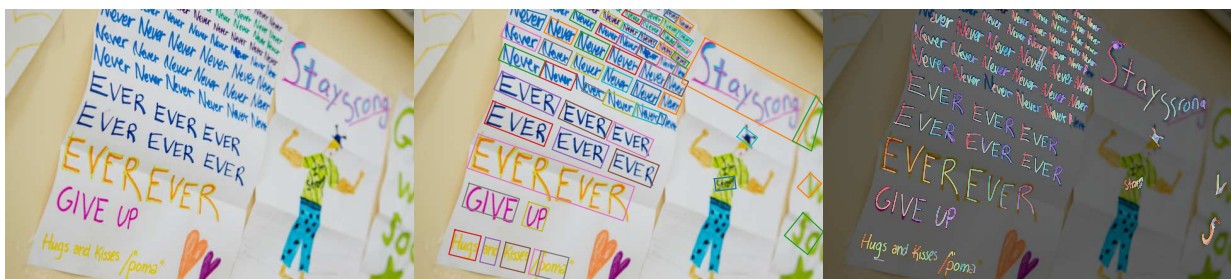

Figure 12: **Small-i** full page results of handwritten notes in a real-life scenario, credit: Unsplash.

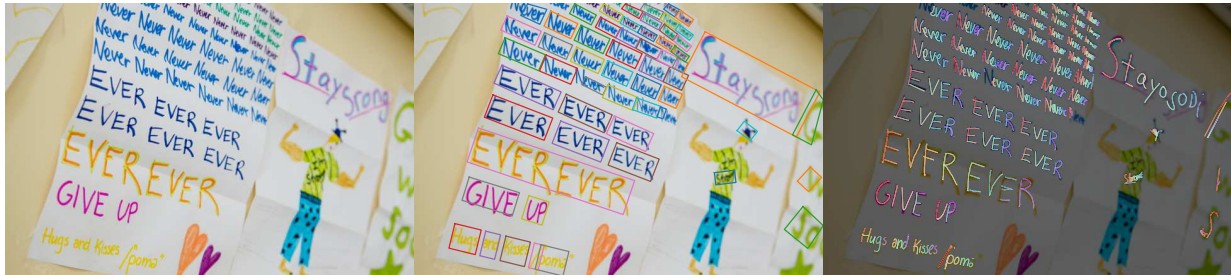

Figure 13: **Small-p** full page results of handwritten notes in a real-life scenario, credit: Unsplash.

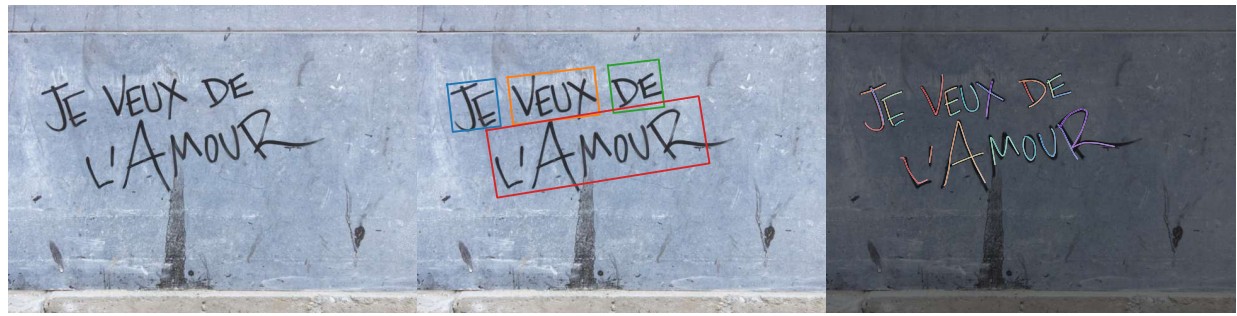

Figure 14: **Large-i** full page results of French sample. credit: Unsplash

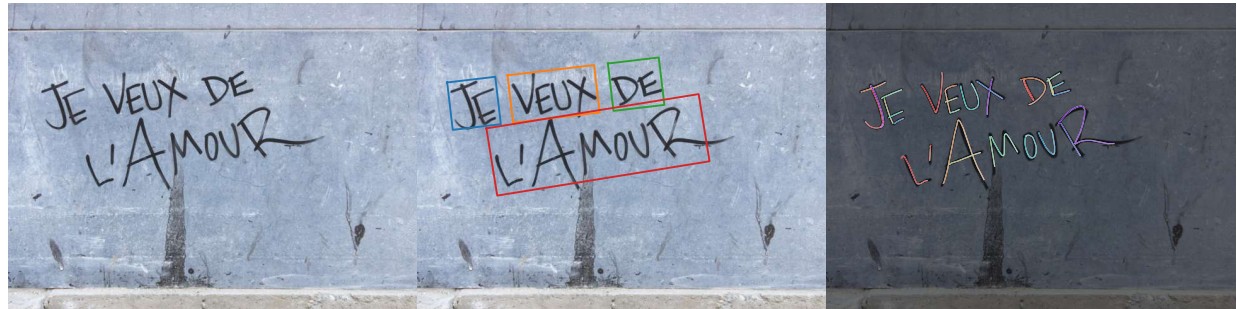

Figure 15: **Small-i** full page results of French sample. credit: Unsplash

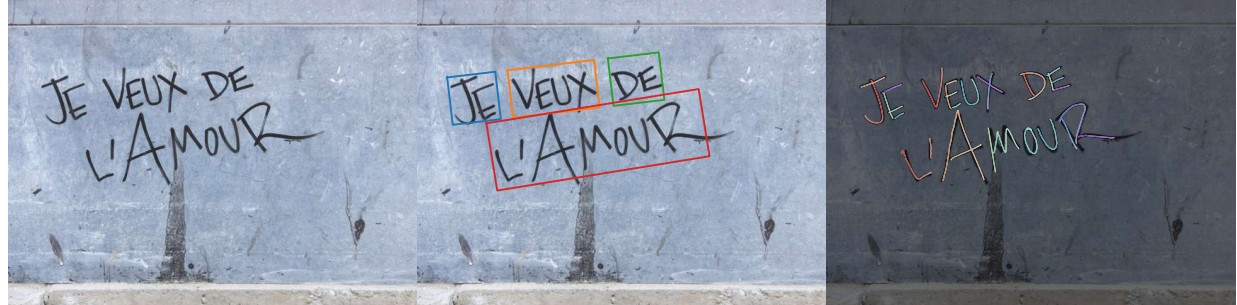

Figure 16: **Small-p** full page results of French sample. credit: Unsplash

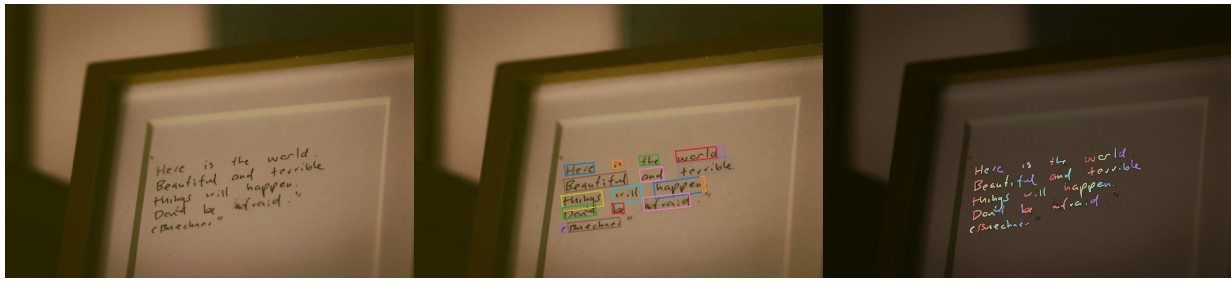

Figure 17: **Large-i** full page results of handwritten notes in a real-life scenario with low resolution, credit: Unsplash.

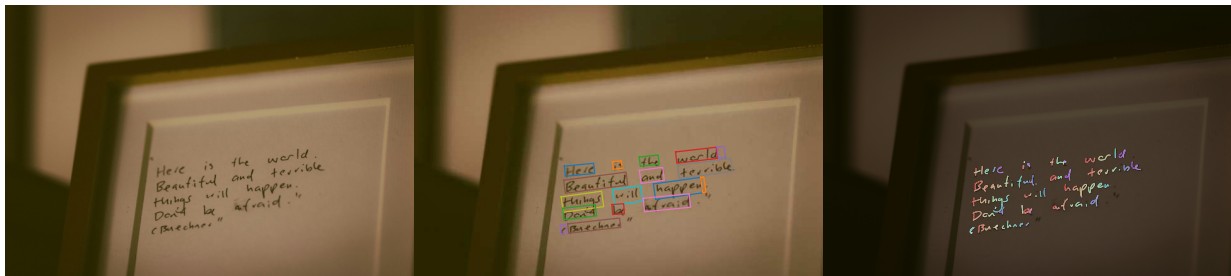

Figure 18: **Small-i** full page results of handwritten notes in a real-life scenario with low resolution, credit: Unsplash.

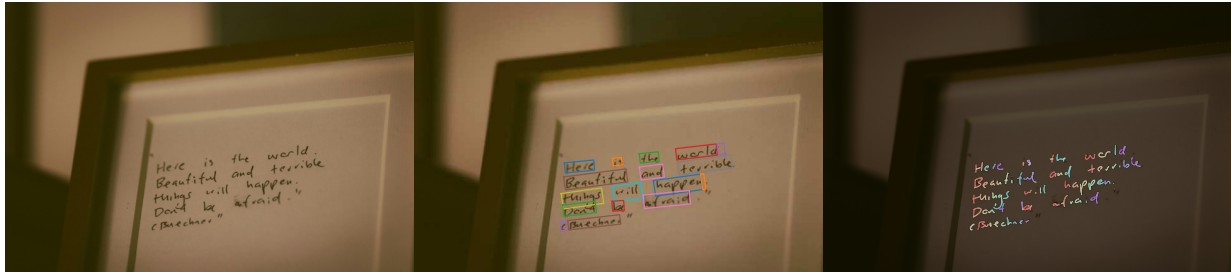

Figure 19: **Small-p** full page results of handwritten notes in a real-life scenario with low resolution, credit: Unsplash.

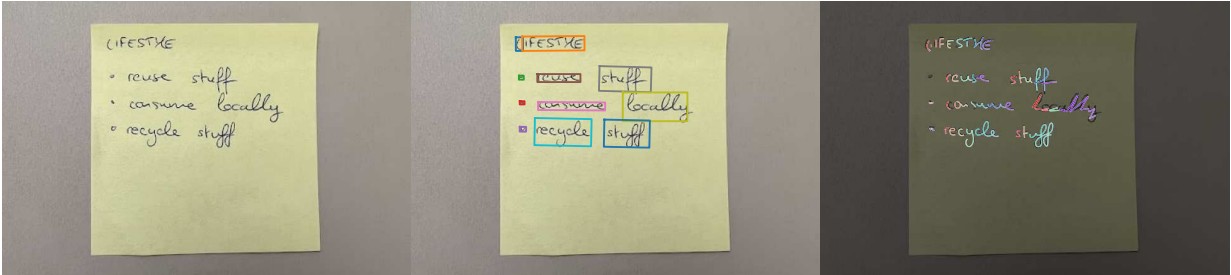

Figure 20: **Large-i** full page results of a Post-It note.

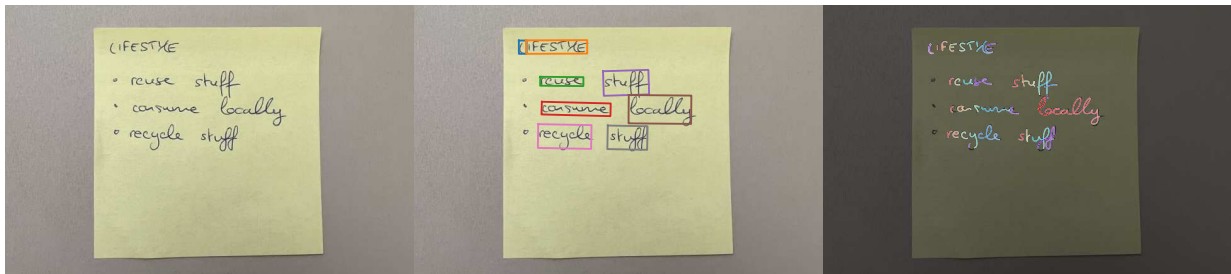

Figure 21: **Small-i** full page results of a Post-It note.

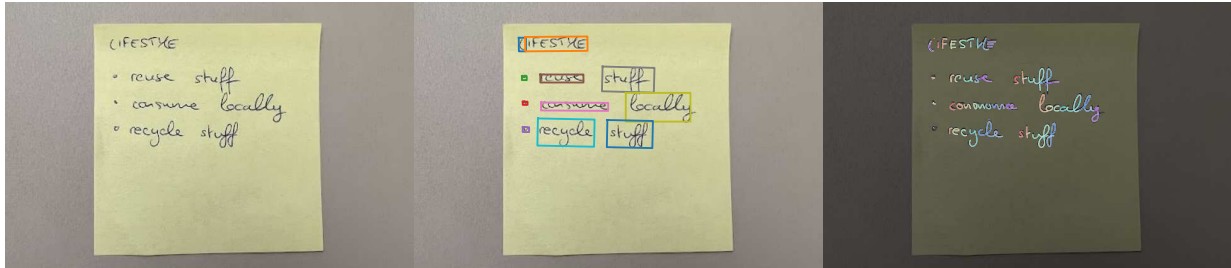

Figure 22: **Small-p** full page results of a Post-It note.

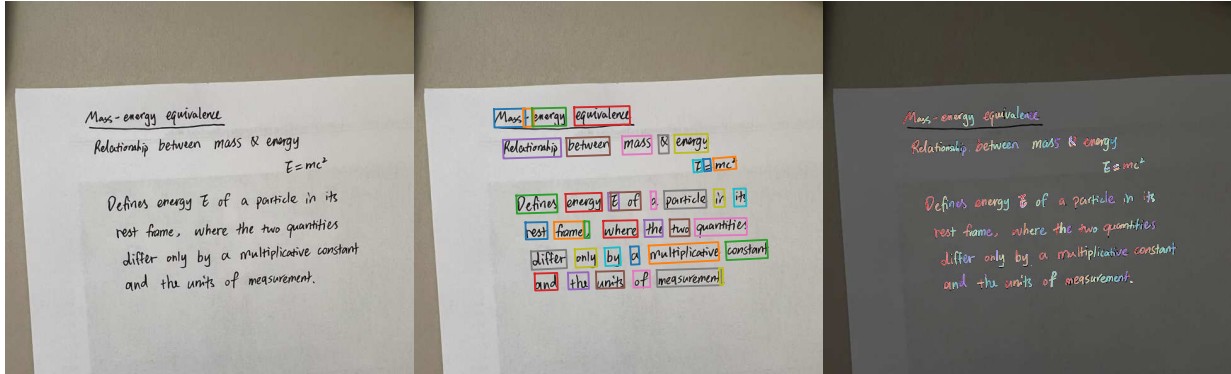

Figure 23: **Large-i** full page results of handwritten notes of mass-energy equivalence.

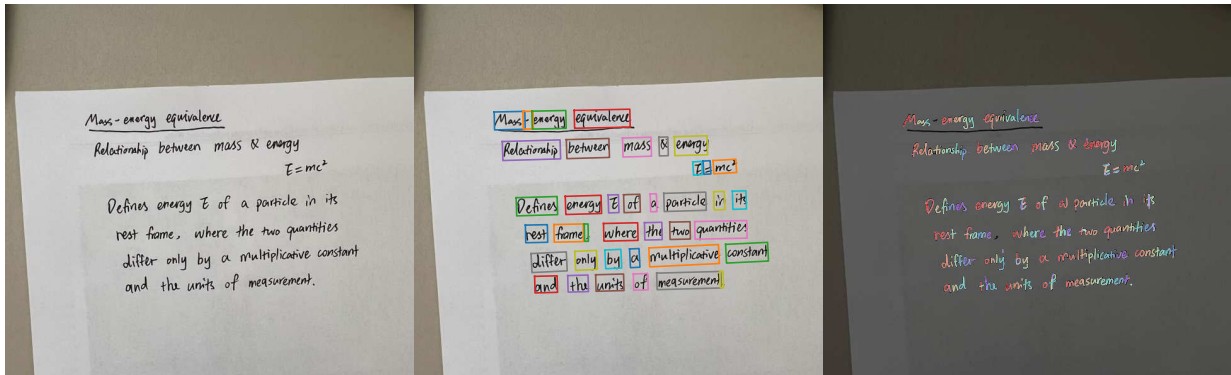

Figure 24: **Small-i** full page results of handwritten notes of mass-energy equivalence.

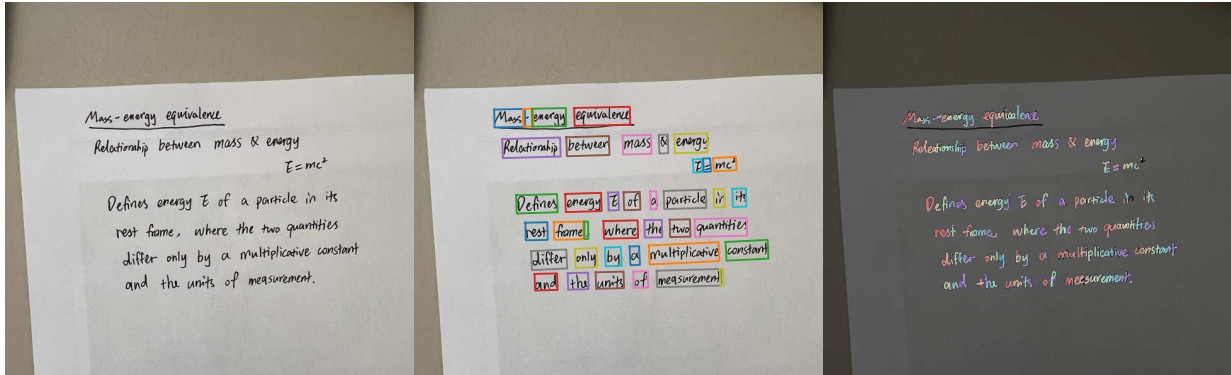

Figure 25: **Small-p** full page results of handwritten notes of mass-energy equivalence.

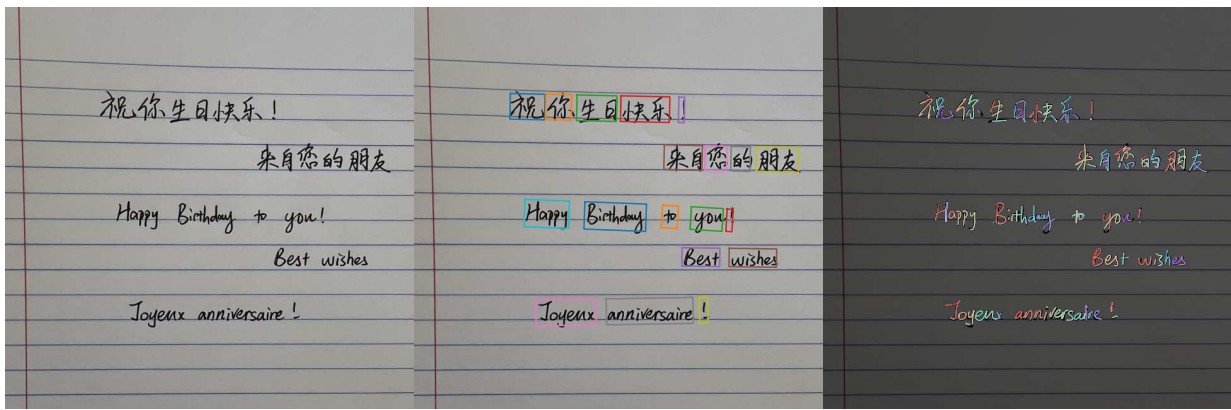

Figure 26: **Large-i** full page results of multilingual handwritten notes.

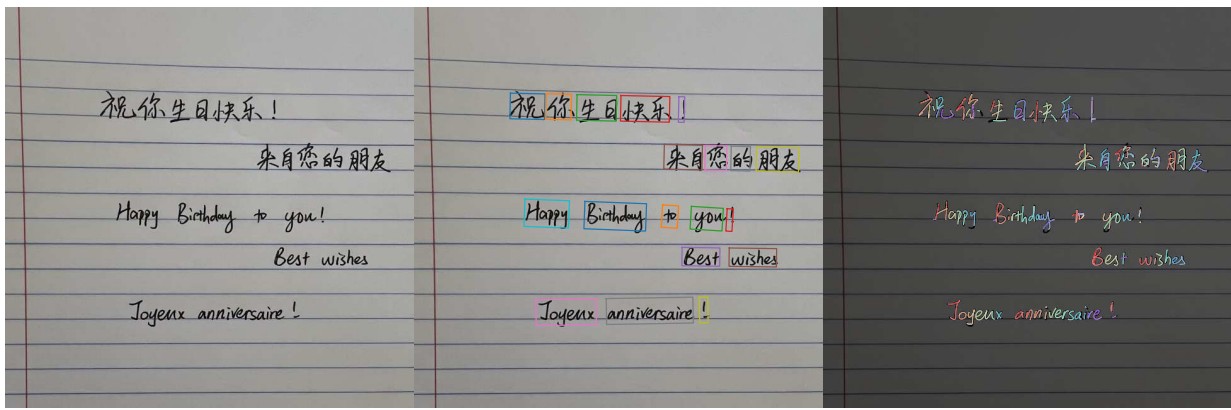

Figure 27: **Small-i** full page results of multilingual handwritten notes.

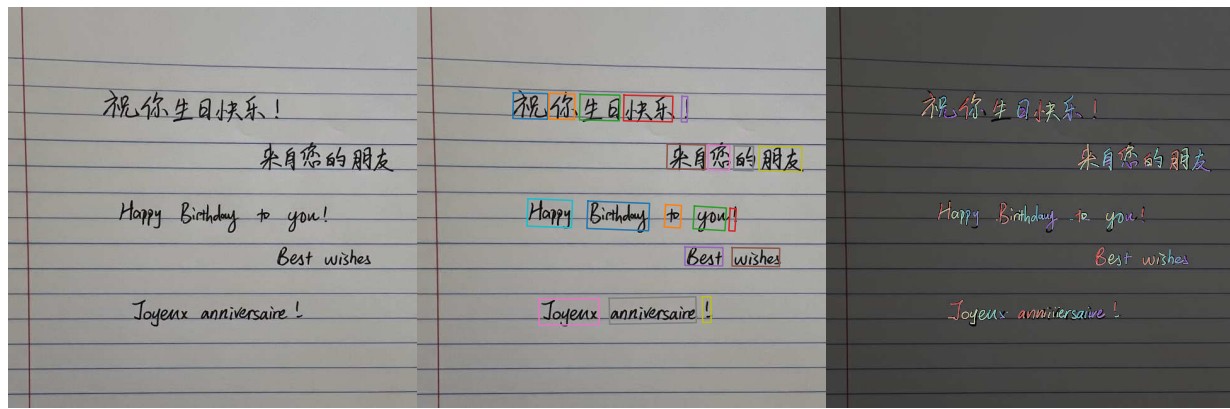

Figure 28: **Small-p** full page results of multilingual handwritten notes.

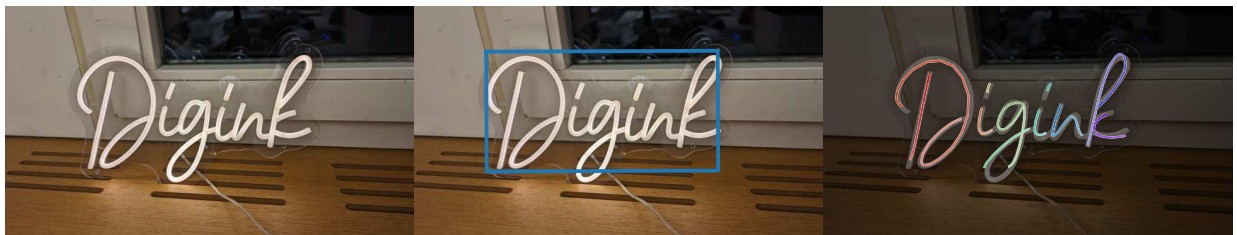

Figure 29: **Large-i** full page results of OOD sample.

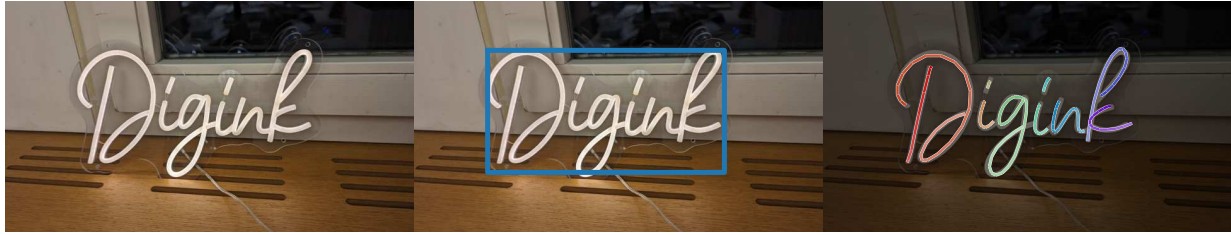

Figure 30: **Small-i** full page results of OOD sample.

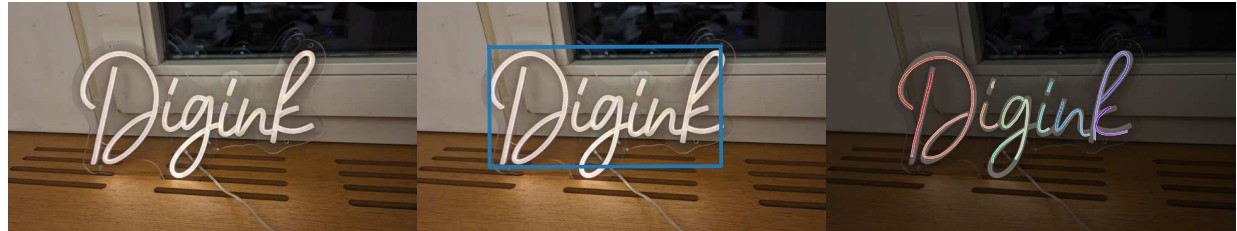

Figure 31: **Small-p** full page results of OOD sample.

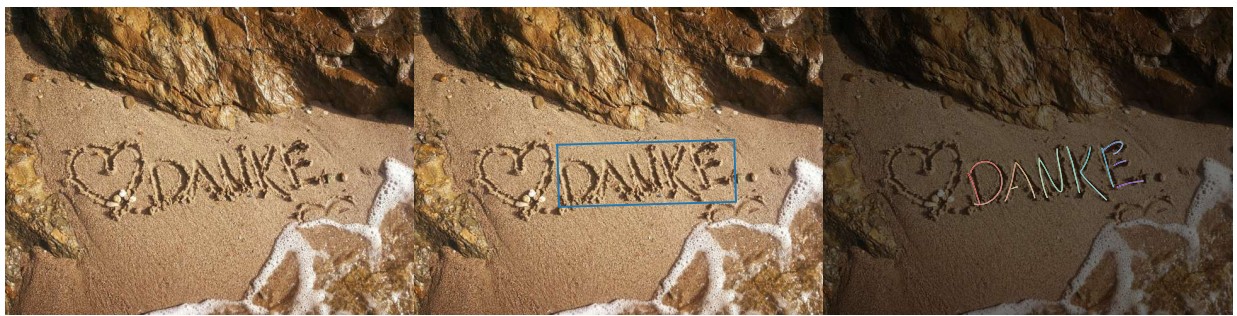

Figure 32: **Large-i** full page results of OOD sample, credit: Unsplash.

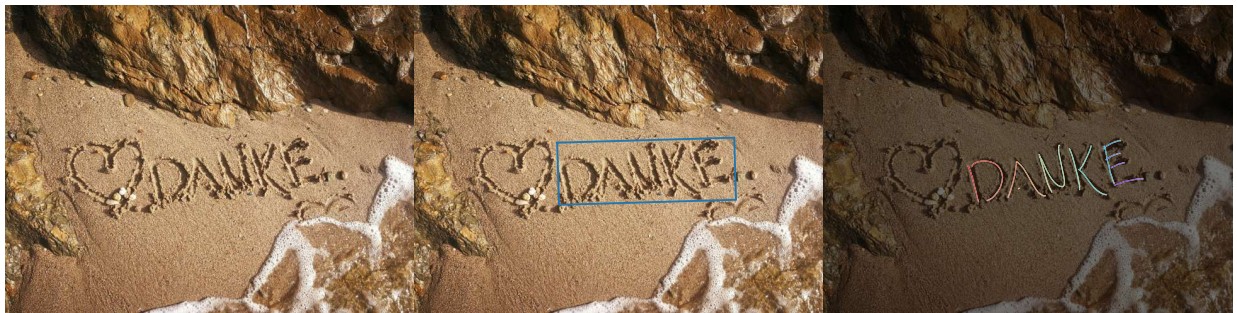

Figure 33: **Small-i** full page results of OOD sample, credit: Unsplash.

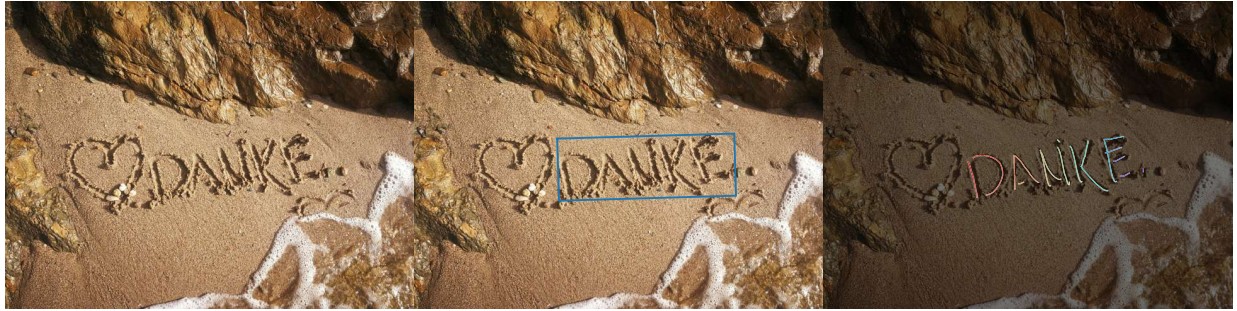

Figure 34: **Small-p** full page results of OOD sample, credit: Unsplash.

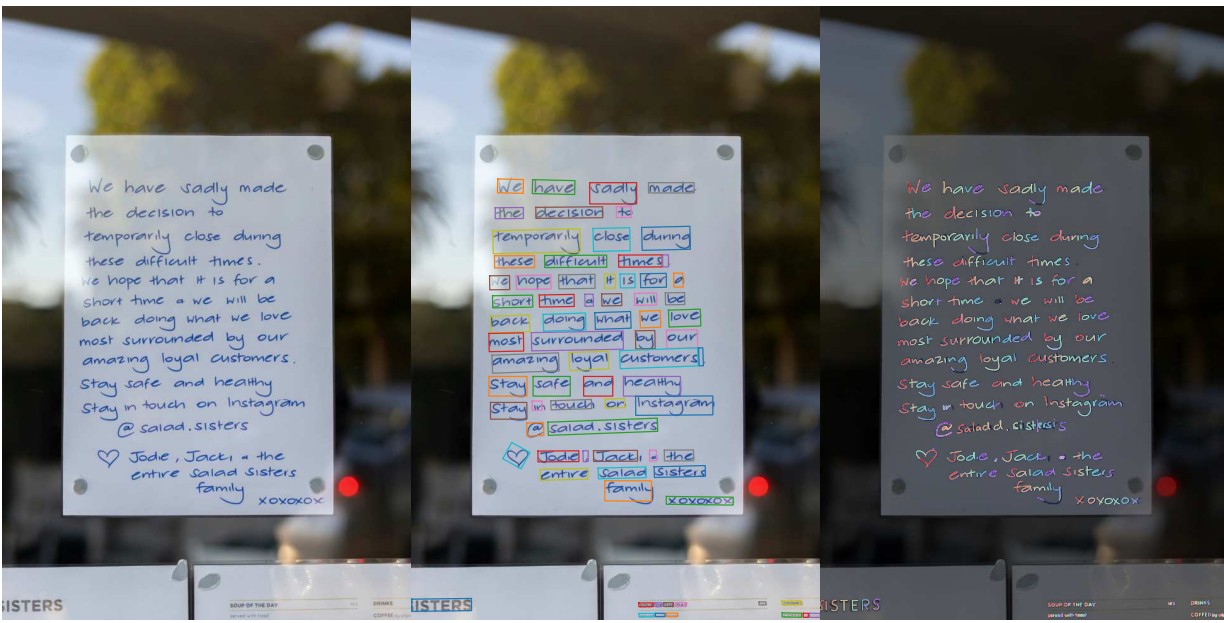

Figure 35: **Large-i** full page results of handwritten notes in a real-life scenario, credit: Unsplash.

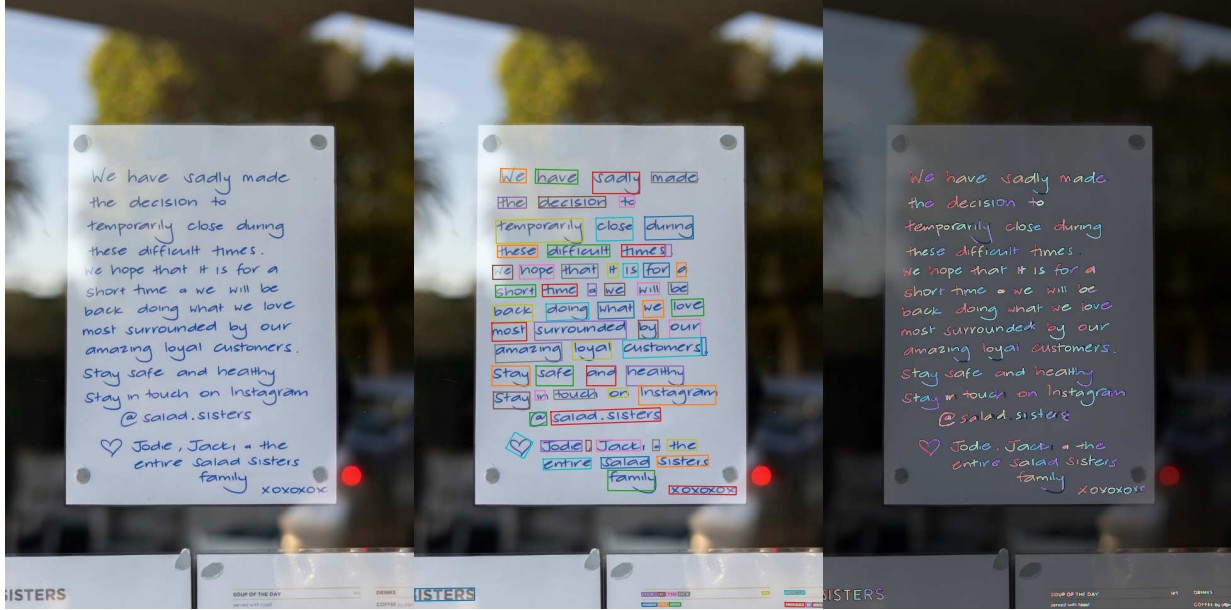

Figure 36: **Small-i** full page results of handwritten notes in a real-life scenario, credit: Unsplash.

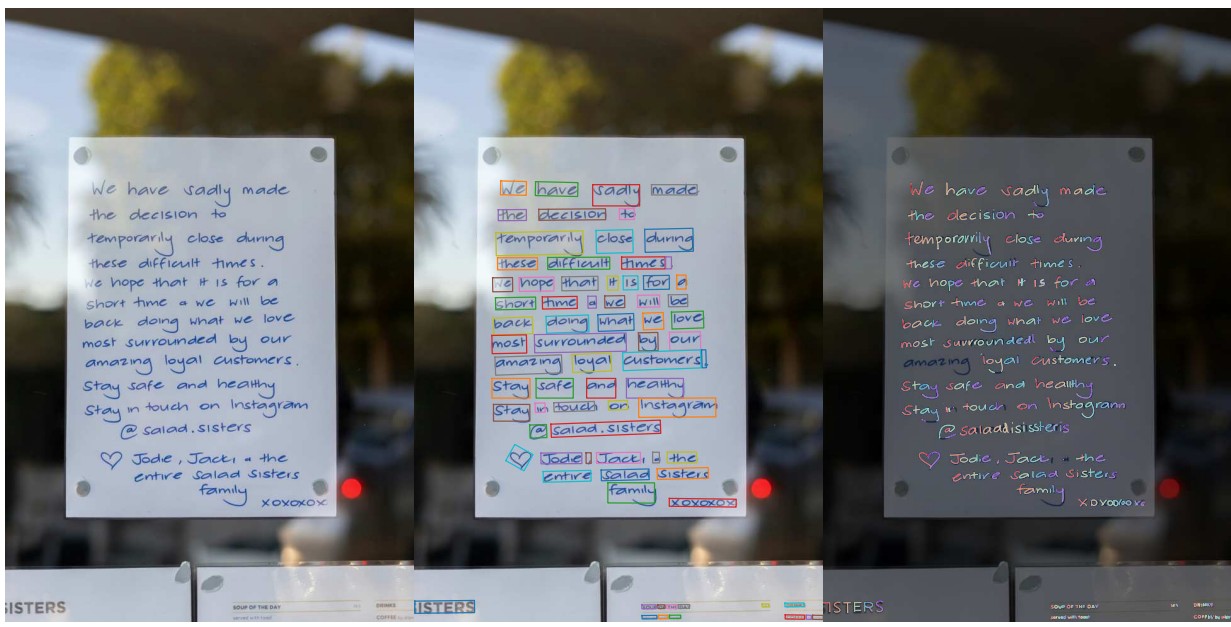

Figure 37: **Small-p** full page results of handwritten notes in a real-life scenario, credit: Unsplash.

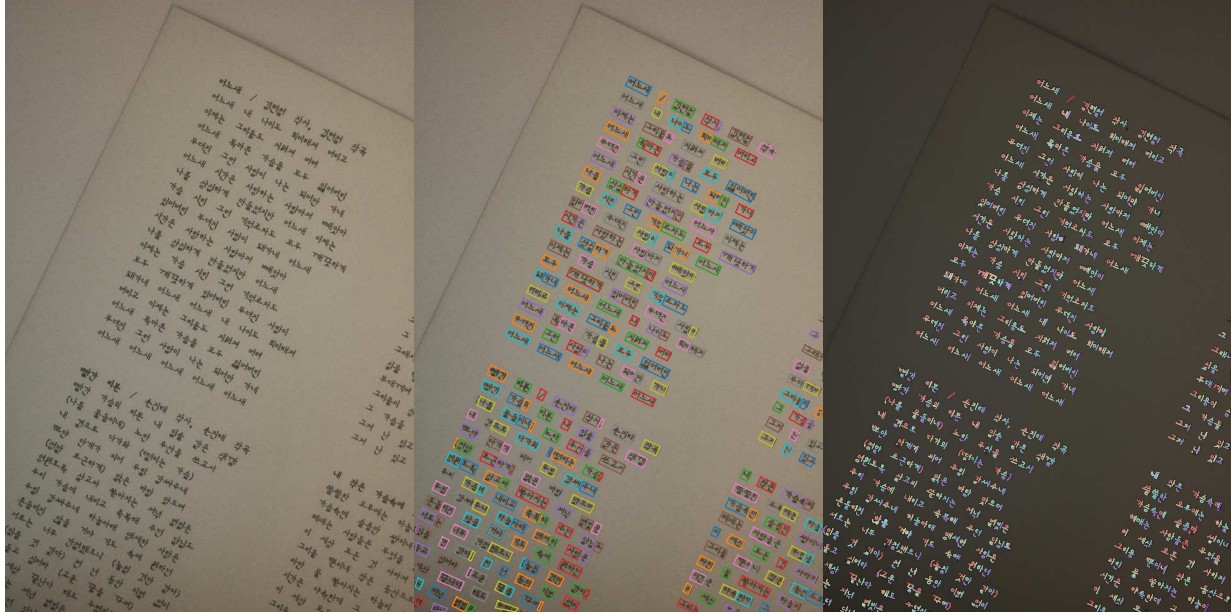

Figure 38: **Large-i** full page results of handwritten notes in a real-life scenario in Korean, credit: Unsplash.

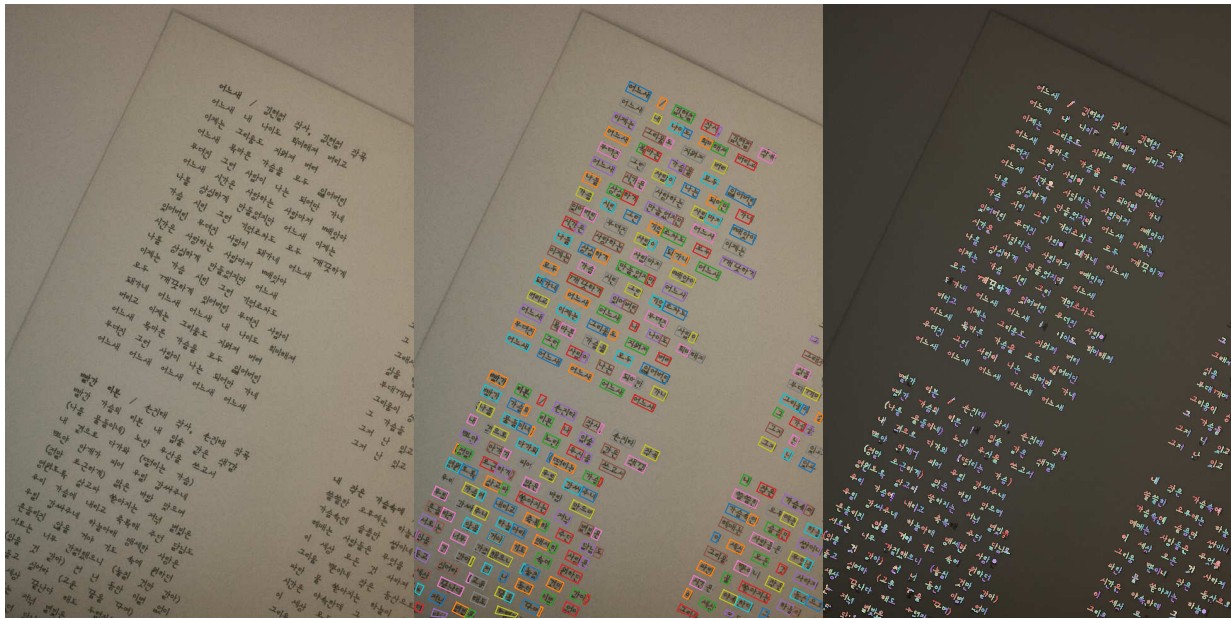

Figure 39: **Small-i** full page results of handwritten notes in a real-life scenario in Korean, credit: Unsplash.

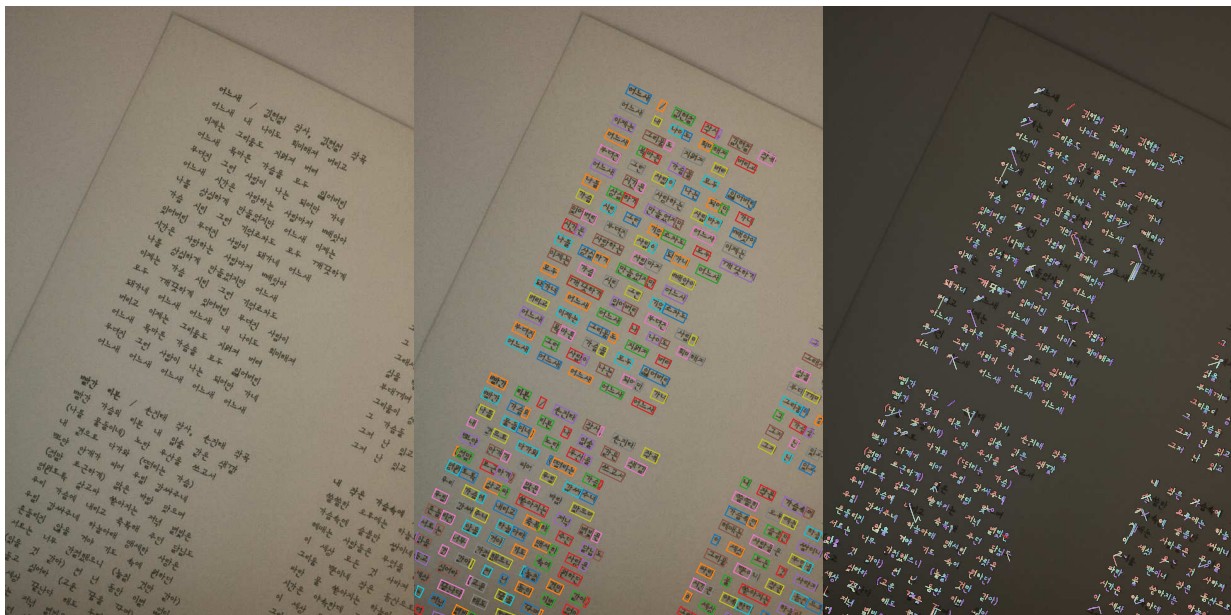

Figure 40: **Small-p** full page results of handwritten notes in a real-life scenario in Korean, credit: Unsplash.

# C    Data statistics and Preprocessing

We present the language distribution of the digital ink datasets used to train our in-house (**-i** suffix) and publicly available (**-p** suffix) models in Figures 41 and 42, datasets and corresponding number of samples used in training our public model in Table 4. The in-house OCR dataset comprises of approximately 500k samples, and in-house Digital Ink dataset comprises of approximately 16M samples.

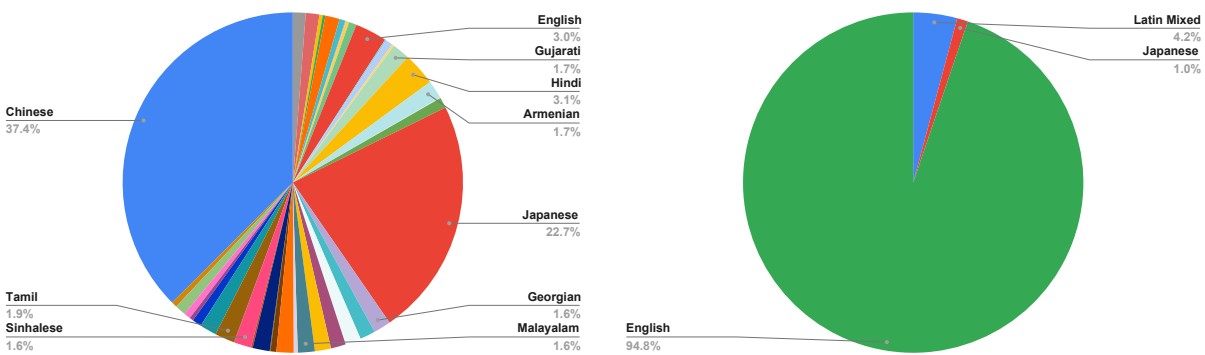

Figure 41: In-house Datasets Language Distributions. **Left:** Digital inks, **Right:** OCR for Recognition. These datasets are used to train both in-house models **Small-i** and **Large-i**.

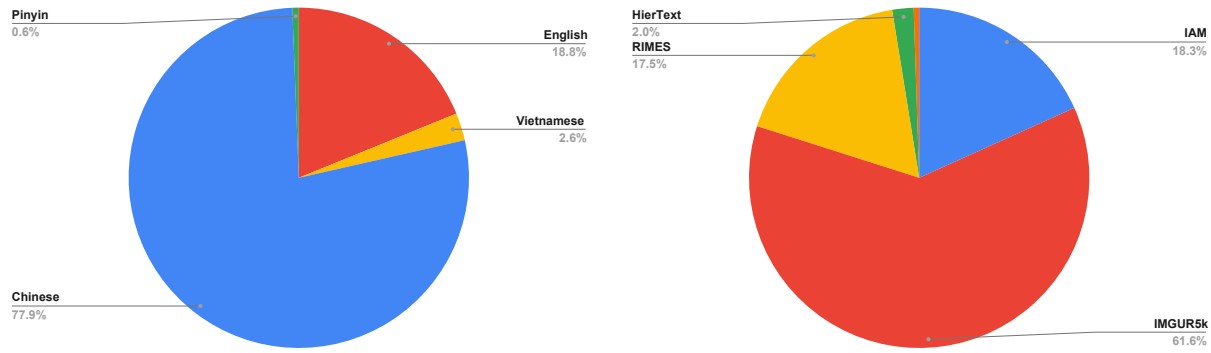

Figure 42: Public Datasets. Language distribution of public Digital ink datasets used to train our public models (on the left) and public OCR datasets used to train our public available model **Small-p** (on the right).

# D    Survey of Derendering Models

In this section we provide a small survey of recent published pen trajectory recovery approaches and show that none of them release sufficient materials to reproduce their work with a reasonable amount of effort. We illustrate this with Table 5.

Table 4: Public training datasets used for training **small-p**.

| Task Type | Dataset | Samples | License |
|---|---|---:|---|
| Derendering | DeepWriting (words) | 89,565 | Custom[1] |
| | DeepWriting (lines) | 33,933 | Custom[1] |
| | DeepWriting (characters) | 359,643 | Custom[1] |
| | VNOnDB | 66,991 | MIT[2] |
| | SCUT-COUCH (chars) | 1,998,784 | Custom[3] |
| | SCUT-COUCH (pinyin) | 15,653 | Custom[3] |
| Recognition | IAM word-level (train) | 53,839 | Custom[4] |
| | IMGUR5k (train) | 181,792 | CC BY-NC[5] |
| | RIMES word-level (train) | 51,738 | CC BY-NC[5] |
| | HierText (train) | 5,978 | CC BY-SA[6] |
| | ICDAR-2015 (train) | 1,535 | Custom[7] |

Custom license (CC BY-NC-SA 4.0 + non-distribution). See: URL.
MIT License. See: URL.
Custom license (research-only, attribution). See: URL.
Custom license (research-only, non-commercial, attribution). See: URL.
CC BY-NC 4.0. See: URL.
CC BY-SA 4.0. See: URL.
Custom license (research-only, attribution). See: URL.

Table 5: Existing pen trajectory recovery approaches and their reproducibility.

| Study | Code Availability | Data Availability | Model Weights Availability | Year |
|---|---|---|---|---|
| Mohamed Moussa et al. (2023) | ✗ ✗ | ✓ | ✗ | 2023 |
| Chen et al. (2022b) | Repo missing core elements: data preprocessing, incompatible configs | ✓ | ✗ | 2022 |
| Archibald et al. (2021) | ✗ ✗ | ✓ | ✗ | 2021 |
| Bhunia et al. (2021) | Repo missing core elements: training scripts, data preprocessing | ✓ | ✗ | 2021 |
| Nguyen et al. (2021) | ✗ | ✓ | ✗ | 2021 |
| Sumi et al. (2019) | ✗ | ✓ | ✗ | 2019 |
| Bhunia et al. (2018) | ✓ | ✗ | ✗ | 2018 |

# E    Further Details of Ablation Studies

## E.1    Frozen ViT helps training stability

We investigated the impact of freezing the vision encoder during training on model stability and derendering performance. To assess training stability, we analyzed the ratio of empty ink predictions across training steps on the "golden" human traced dataset (detailed in Section 4.5.1). Empty inks typically occur when the model confuses tasks, mistakenly outputting text instead of ink or both (different tasks depicted in Figure 3). Our analysis revealed significantly greater variance between runs with an unfrozen setup compared to the consistent learning observed with a frozen vision encoder (Figure 43).

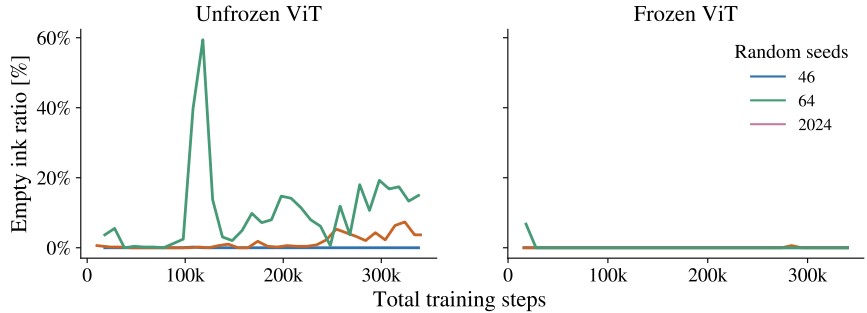

Figure 43: **Comparison between frozen and unfrozen ViT Small-i trainings.** Comparison is executed with 3 random seeds. **Left:** ratio of empty ink outputs for unfrozen ViT multi-task training setup. **Right:** ratio of empty ink outputs for the frozen ViT multi-task training setup, both use inference task *Derender with Text.*

Furthermore, in terms of final model derendering performance, we identified that certain model initializations (controlled by seeds) within the unfrozen setup could also converge to functional models with comparable abilities to those trained with a frozen setup. These models demonstrate superior preservation of visual ink details but exhibit increased sensitivity to background noise. We present a visual comparison of their output characteristics in Figure 44.

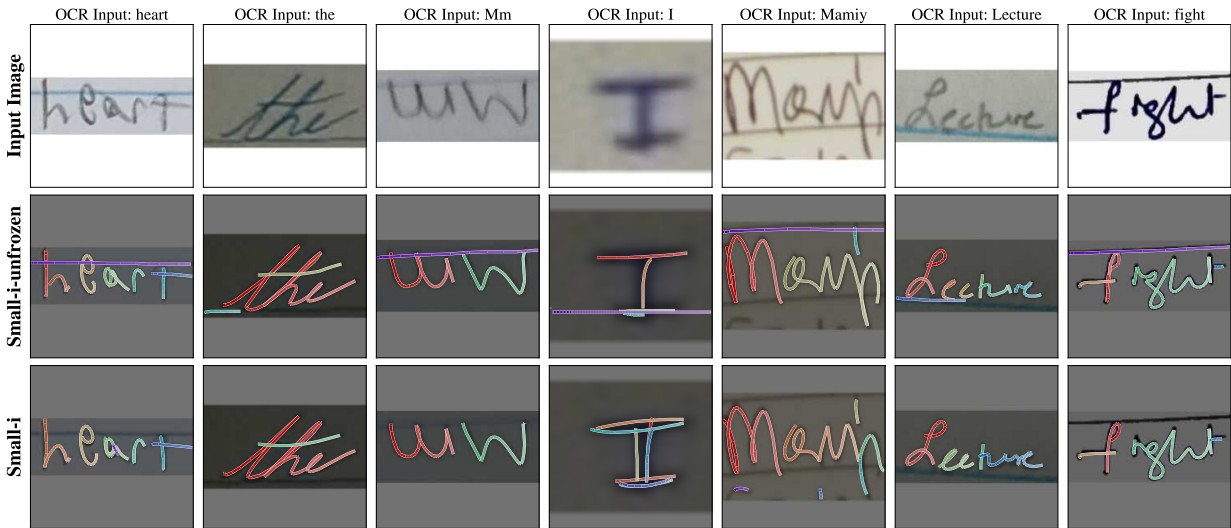

Figure 44: Difference between **Small-i-unfrozen** which was trained with an unfrozen ViT (seed 46 in Figure 43) and **Small-i** which was trained with a frozen ViT. This shows that **Small-i-unfrozen** is more sensitive to background noise compared to **Small-i**.

## E.2 Inference type matters

As we first illustrated in Section 4.6, different inference tasks can produce different output digital inks especially when the input image of text is semantically or geometrically ambiguous due to either image quality or how the texts were written. For example the first column of Figure 45 where the word "wich" is removed by the writer with a strike through line but still recognized by both *Recognize and Derender* (model intrinsic) and *Derender with Text* (extrinsic OCR system) inference, and the difference in understanding and locating the textual information in input image results in different output digital inks. Additionally, the output from *Vanilla Derender* where semantics are not intentionally involved during training is more robust to ambiguous inputs but shows poorer semantic consistency with the input image.

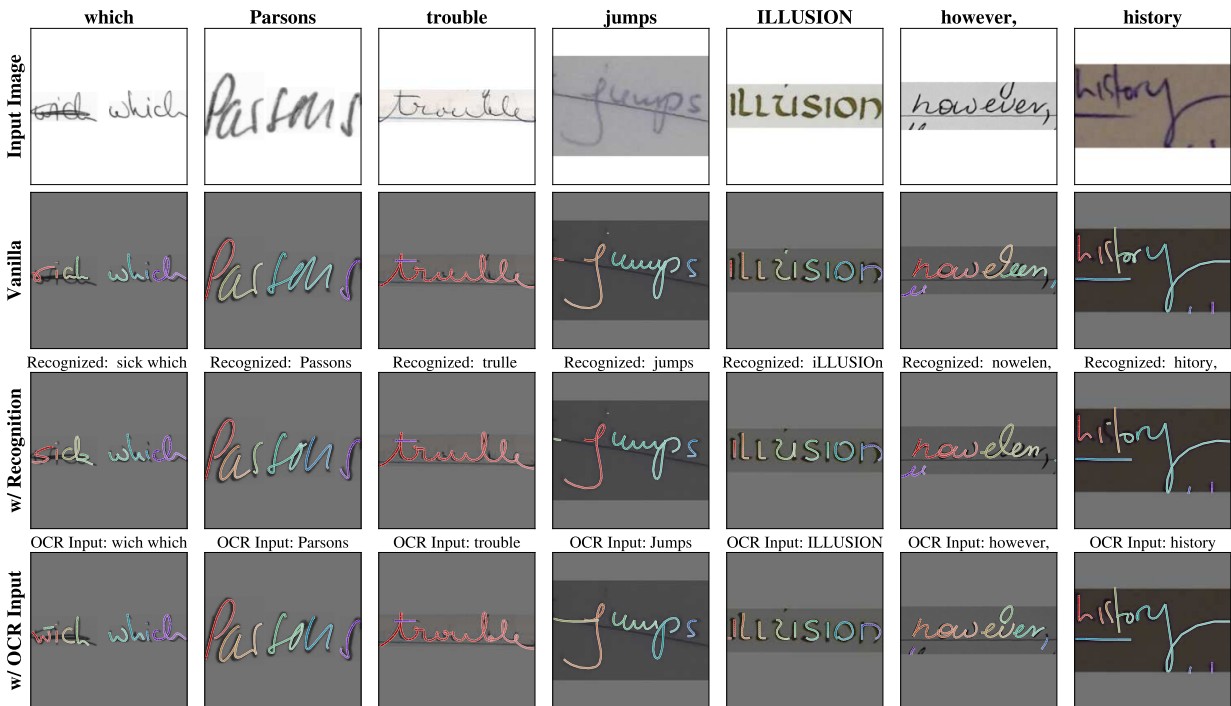

Figure 45: **Difference between inference tasks** of **Small-i** on samples from 3 public evaluation datasets, where the texts on top are the ground truth labels for the images in these datasets. **Vanilla** stands for inference with *Vanilla Derendering*, **w/Recognition** stands for inference with *Recognize and Derender*, and **w/ OCR Input** stands for inference with *Derender with Text* where we resort to an OCR system.

For *Derender with Text* inference, we have shown that how external textual input could benefit the model's semantic understanding. We explore the behavior of the model in cases where the external text is different than the ground-truth textual information in Figure 46. The results reveal that the model's output is not solely determined by input text. It demonstrates the ability to produce valid outputs even when external textual information is incomplete or erroneous.

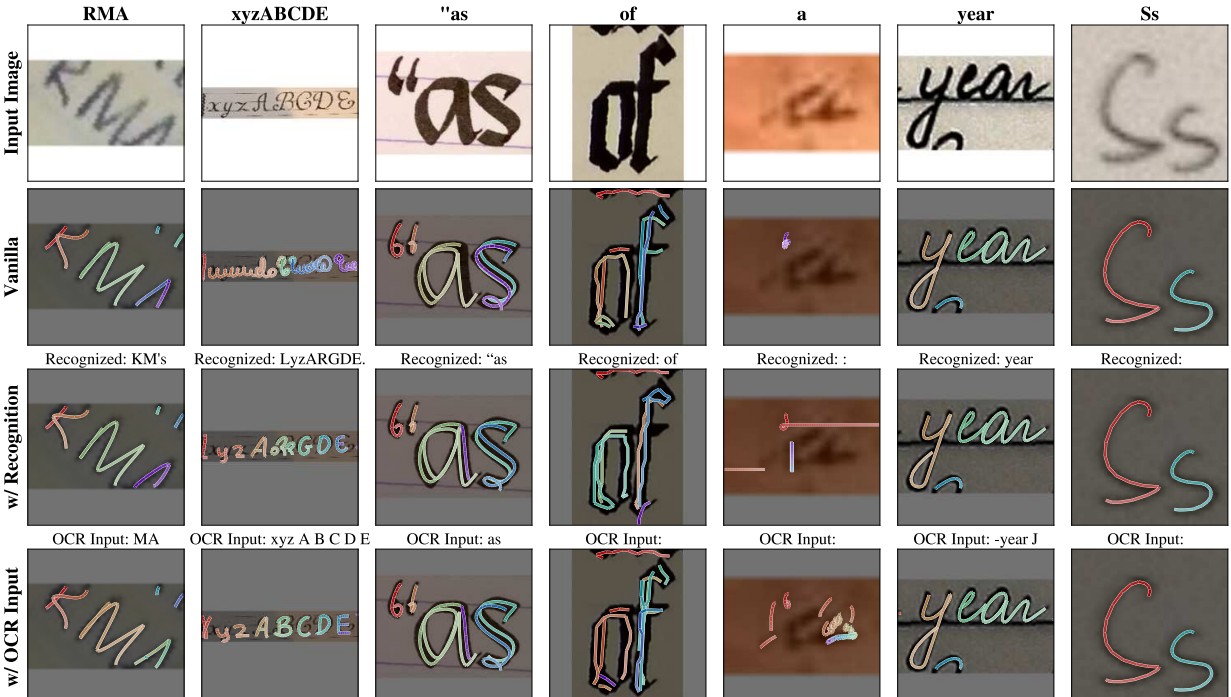

Figure 46: **Cases when there is a mismatch between ground-truth textual information and external textual information provided to the model.** We evaluated this using **Small-i** on samples from 3 public evaluation datasets, where the texts on top are the ground truth labels for the images in these datasets. **Vanilla** stands for inference with *Vanilla Derender*, **w/Recognition** stands for inference with *Recognize and Derender*, and **w/ OCR Input** stands for inference with *Derender with Text* where we resort to an OCR system.

## F   Rendering and Data Augmentation

To create synthetic rendered inks of our online samples, we use the Cairo graphics library to render online ink samples onto a $224 \times 224$ canvas. We then add several augmentations to rendered samples.

Before rendering, we first randomly rotate the samples. Then, we pick the color of the strokes and the background uniformly at random from the RGB color space and render the ink with a random width. Furthermore, we add lines, grids, or Gaussian noise into the background with a fixed probability. Finally, we potentially add box blur on the resulting image. This approach was mainly inspired by approaches from synthetic OCR generation, e.g. as described in (Etter et al., 2019). The detailed parameters of data augmentation and some samples are shown in Table 6. An example of how samples augmented with all parameters chosen at random could appear in the training set can be seen in Figure 47. Other approaches, such as perspective skew, shifting, scaling, or padding, did not show any improvements in the model performance and were discarded.

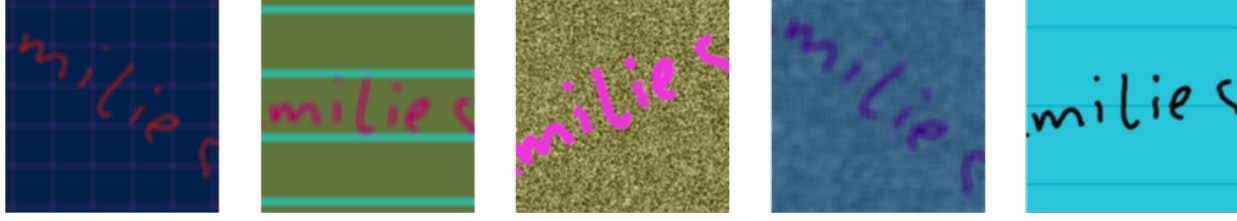

Figure 47: Demonstration of possible ways to combine all used augmentations. (Sample from (Aksan et al., 2018))

Table 6: Data Augmentations Overview (sample from Aksan et al. 2018).

| Augmentation | Possible Values | Examples |
|---|---|---|
| Rotation | angle (rad): $[-\frac{\pi}{4}, \frac{\pi}{4}]$ | |
| Stroke color | RGB: $[0,1]^3$ | |
| Background color | RGB: $[0,1]^3$ | |
| Stroke Width | width: $[1px, 12px]$ | |
| Lines | line width: $[1px, 6px]$ line dist: $[10px, 100px]$ line color: RGB: $[0,1]^3$ | |
| Grids | line width: $[1px, 6px]$ line dist: $[10px, 100px]$ line color: RGB: $[0,1]^3$ | |
| Gaussian Noise | standard dev.: $[50, 500]$ | |
| Box blur | radius: $[0px, 5px]$ | |

# G    Additional Model Details

The overview and individual components of three versions of our model **Small-i**, **Small-p**, and **Large-i** is shown in Table 7.

Table 7: Model overview and components.

| Model | Components | Parameters | Training Data |
|---|---|---|---|
| **Large-i** | ViT-L
mT5-Large | 303M
783M | In-house proprietary datasets
as shown in Appendix C. |
| **Small-i** | ViT-B
mT5-Base | 85M
247M | In-house proprietary datasets
as shown in Appendix C. |
| **Small-p** | ViT-B
mT5-Base | 85M
247M | Publicly available datasets
as shown in Appendix C. |

**Implementation Details.**    Similar to PaLI models (Chen et al., 2022a; 2023a;b), our models together with the training mixtures are implemented in `JAX/Flax` (Bradbury et al., 2018) using the open-source `T5X`, `SeqIO` (Roberts et al., 2023) and `Flaxformer` (Heek et al., 2023) frameworks. For training, we use the Adafactor optimizer (Shazeer & Stern, 2018) with $\beta_1 = 0$, second order moment of 0.8, and a language-model–style teacher forcing with softmax cross-entropy loss. For the learning rate schedule, we use the linear decay scheduler with a peak learning rate of 0.01, 5k steps of warmup and a linear decay with decay factor of $3e - 6$, and a dropout of 0.1 on both ViT encoder and the mT5 encoder-decoder. We train our models for 340k steps with batch size 512. With frozen ViT encoders, the training of a 340M parameter model (such as **Small-i** or **Small-p**) takes ∼33h on 64 TPU v5e chips and the training of a 1B parameter **Large-i** model takes ∼105h on 64 TPU v5e chips, both run on an internal TPU v5e cluster and should be reproducible with public GCP TPU v5e. Inference with a 340M parameter model takes around 1 hour for all 3 datasets, and 3 hours with **Large-i**.

**Compute.**    For the results reported in Table 3, we train one 340M parameter model per ablation. For entries with estimated variances, we train each model with 3 different seeds, respectively. This totals in ∼330h on 64 TPU v5e chips of training. We additionally train **Small-p** and **Large-i** to obtain the results in Table 2, accounting for ∼138h on 64 TPU v5e chips. In total, all experiments in this paper require a total of ∼500h of compute for training the models and running inference.

**Decoding.**    In our experiments, we observe marginal performance improvements in derendering tasks through adjustments in temperature sampling. However, for the sake of reproducibility and framework simplicity, we employ a greedy decoding strategy during inference for all tasks, with model checkpoints selected from the final step.

**Training Mixture.**    As described in Section 3.2, the training mixture consists of 5 tasks constructed using SeqIO (Roberts et al., 2023). These tasks are pre-shuffled before training, ensuring each task appears with equal probability throughout the entire training process. And to foster reproducibility, the input pipeline as well as the model initialization are designed to be deterministic, reducing randomness-induced variations that could affect our design choices.

**Inference Speed.**    We benchmarked inference speed on three different GPU devices by measuring tokens processed per second over five runs. The Titan RTX achieved an average of 114.43 tokens/s, the T4 averaged 47.14 tokens/s, and the V100 averaged 139.14 tokens/s. All measurements were conducted using greedy decoding under consistent conditions with batch size 32 across devices.

## H    General Virtual Sketching Framework Performance

We demonstrate the performance of three GVS models from the official GitHub repository: Line Drawing Vectorization, Rough Sketch Simplification, and Photo to Line Drawing Conversion in two setups (original image as input and binarized image as input) in Figure 48. We inspect the samples visually on the more challenging HierText dataset. The photo to line drawing model was trained on people portrait photos and fails to generalize to the images of text. Binarization is beneficial for both remaining setups and reduces the noise when derendering texts. Line Drawing Vectorization and Sketch Simplification perform similarly on the inspected samples, we choose Line Drawing Vectorization on binarized images as a baseline since this model was trained on black & white raster images and therefore treats binarized samples as in-domain. For derendering of the sketches (Figure 7) we use the Sketch Simplification model as intended by the authors.

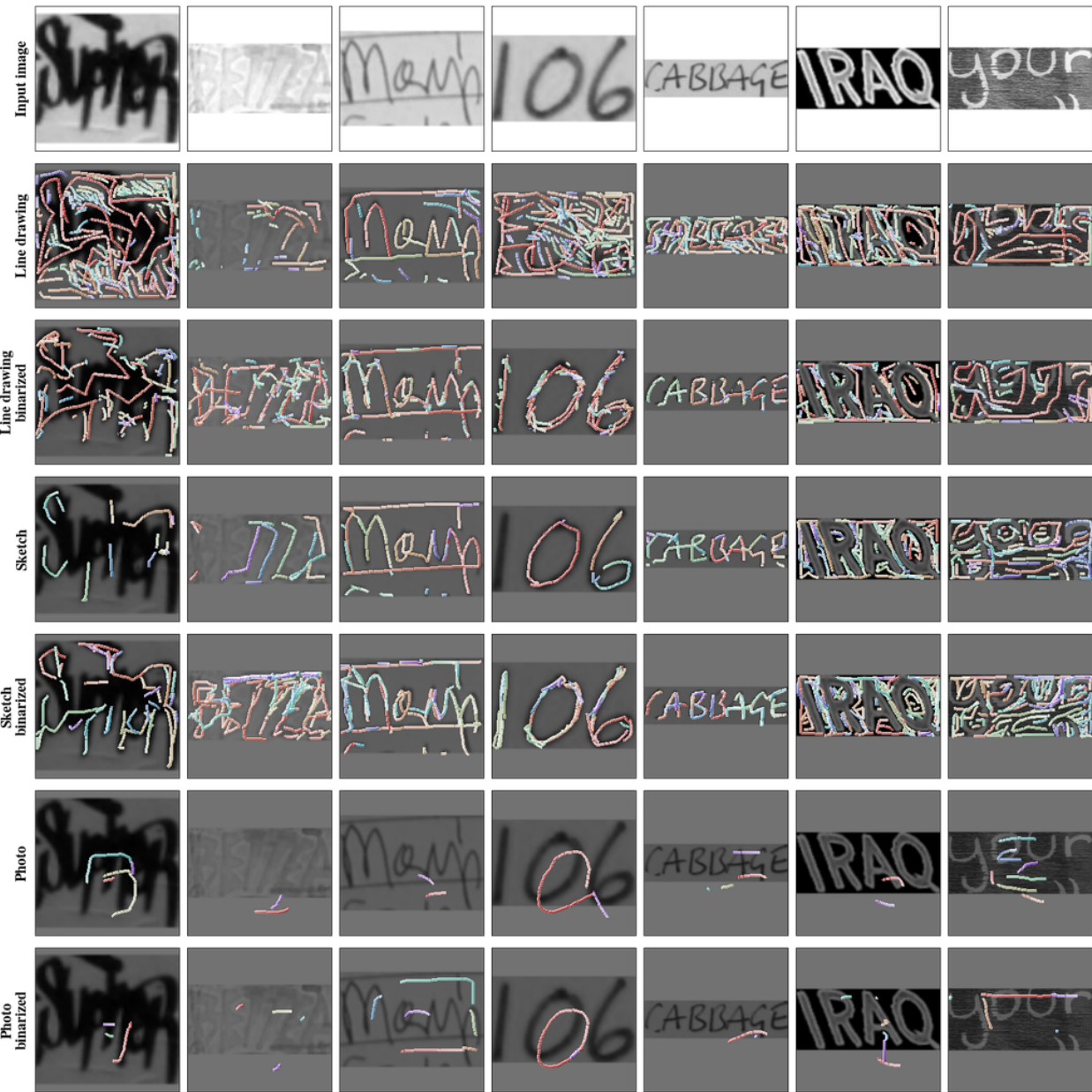

Figure 48: Comparison between General Virtual Sketching Framework inference setups.

# I   Online Handwriting Recognizer Details

In this section we provide the result of online handwriting recognizers used to evaluate the low-level feature similarity of real inks for IAMOnDB dataset and inks derendered by **Small-p**, **Small-i** and **Large-i** (see Table 8). We observe that the inks acquired with all of our models perform similarly when used as training data both on their own and in combination with real digital inks.

Table 8: Online handwriting recognition results on IAMOnDB test set for models trained on real digital inks (IAMOnDB train set), derendered digital inks (from IAM train set) and the combination of the two.

|  | IAMOnDB | Small-i IAM derendered | Small-i IAMOnDB + IAM derendered | Small-p IAM derendered | Small-p IAMOnDB + IAM derendered | Large-i IAM derendered | Large-i IAMOnDB + IAM derendered |
|---|---|---|---|---|---|---|---|
| CER | 6.1 | 7.8 | 4.6 | 8.2 | **4.5** | 8.4 | 4.6 |

Additionally, we describe the architecture and our recognition training procedure. We choose a common approach of training a (relatively small, 9.2M, due to the amount of the training data) Transformer encoder combined with a CTC loss, similar to Alwajih et al. 2022. We fix the same preprocessing steps for both real and derendered inks. Those include shifting and scaling the inks to start at the origin and have a fixed y side, time resampling at 20 ms for real inks and time hallucination (assuming the predicted ink tokens are produced at a fixed sampling rate of 20ms, matching the training data) and adding pen up strokes. We apply a random rotation within $\pm 45°$ with probability 0.5 as data augmentation to account for the small size of the dataset. The points from the ink are encoded into Bezier curves. The input features are then processed by 7 transformer attention layers with 8 attention heads.

We train the models on a training set of IAMOnDB only (baseline), derendered IAM train set only and the combination of the two for 17.5k steps with batch size of 64. We perform a hyper-parameter search for dropout and learning rate on the same grid for all 3 setups. The best parameters for each model can be found in Table 9.

Table 9: Recognizer parameters found through hyper-parameter search.

| Parameter | IAMOnDB | Small-i IAM derendered | Small-i IAMOnDB + IAM derendered | Small-p IAM derendered | Small-p IAMOnDB + IAM derendered | Large-i IAM derendered | Large-i IAMOnDB + IAM derendered |
|---|---|---|---|---|---|---|---|
| Dropout | 0.25 | 0.25 | 0.3 | 0.3 | 0.25 | 0.25 | 0.25 |
| Learning rate | 0.005 | 0.001 | 0.001 | 0.005 | 0.005 | 0.001 | 0.001 |

## J   Expansion on Failure Cases

While our system exhibits satisfactory performance across various tested inputs, we have identified specific failure patterns that warrant further attention.

**Extreme Stroke Width/Style Variations**   As observed in Figures 44 and 46, performance can degrade when processing samples with stroke widths or styles significantly outside the training distribution. While training with our data augmentation improves robustness, it does not fully cover extreme cases. Enhancing robustness to diverse and extreme stylistic variations such as through the inclusion of harder examples in the training mixture or the use of expert models, remains an area for future work.

**Disproportionate Component Scaling**   While scaling the vision (ViT) component enhances detail preservation (Figure 5, Section 4.5.1), we observed that semantic understanding for tasks requiring recognition can be negatively impacted if the text-ink (mT5) component is not scaled proportionally. Finding the optimal balance between visual feature extraction and sequence modeling capabilities during scaling is an important consideration.

**Vision Head Training Trade-offs**   As detailed in Section 4.6, unfreezing the ViT heads allows capturing finer image details but introduces training instability and increased confusion between text and background noise. Future work could explore more sophisticated fine-tuning strategies, potentially adapting techniques like (Zhai et al., 2022) to balance detail capture with stability and semantic accuracy.

**Untrained Languages**   When presented with handwriting in a language completely absent from its training data (as illustrated with the Korean example for Small-p in Figure 40), the model cannot leverage semantic understanding. In these scenarios, it defaults to derendering based primarily on visual features and learned writing dynamics, which can lead to less accurate stroke reconstruction and an inability to utilize text-conditioned generation effectively. This highlights a limitation in applying text-conditioned modes to truly unseen scripts without further adaptation or fine-tuning.

**Task Confusion during Decoding**   Our multi-task training setup, while crucial for injecting priors, can occasionally lead to task confusion during decoding. We observed instances, particularly with the Recognize and Derender task, where the model is prompted to perform semantic understanding might mistakenly prioritize the recognition sub-task and fail to generate the corresponding ink sequence (we use greedy decoding), resulting in an empty ink output. This suggests potential ambiguity in the model's internal representation or decoding process when asked to perform both tasks simultaneously. To mitigate this during inference, we implemented a fallback mechanism: if the Recognize and Derender task produces an empty ink output, we automatically re-run inference using the Vanilla Derender mode to ensure an ink sequence is generated based solely on the visual input. While effective as a practical workaround, addressing the root cause of this task confusion within the model architecture or decoding strategy represents an area for future investigation.

## K  Expansion on Out-of-Domain Generalization

In addition to the Generalization to Sketches as we described for Figure 7, we analyze more evidence of Out-of-Domain Generalization that we see in selected samples.

**Generalization to untrained scripts and languages**   As noted in Appendix J, even though the model was trained only on Chinese, English, and Vietnamese, it can still partially derender Korean text. A similar extension to German is shown in Figure 34, though in both cases the derendering quality is noticeably worse—especially when comparing Small-p with Small-i/Large-i. Because Small-i/Large-i was trained on both languages while Small-p was not, these examples highlight the model's limited generalization ability to untrained scripts.

**Generalization to unseen textures and backgrounds**   We present additional evidence for generalization on a variety of textures and backgrounds, such as a neon sign on a window ledge in Figure 31 or handwriting written in sand in Figure 34.

## L  Human Evaluation

We provide the text of our evaluation instructions in Table 10 and the screenshot of the interface in Figure 49. Furthermore, we show the samples and corresponding rating (after majority voting) in Figure 50.

Table 10: Human evaluation campaign instructions.

| Description | Instruction |
| --- | --- |
| Task setup | You will be looking at input images of photos of handwriting in a variety of styles and the corresponding digital ink tracing performed either by a human or by one of our derendering models.
Each stroke is rendered in a different color and with color gradient (darker → whiter) to reflect the direction.
You will see the derendered ink as well as the derendered ink overlaid on top of the image and can use either to make the judgment.
For each sample answer the two questions on the right and click submit to proceed to the next sample. |
| Is Reasonable | Is the digital ink a reasonable tracing of the input image?

• Yes, it's a good tracing
• It's an okay tracing, but has some small errors
• It's a bad tracing, has some major artifacts

One interpretation of the quality of the ink could be like this:
If you were to use it in a note taking app, would you

• keep it as is: good tracing
• need to make some edits: okay tracing
• not use it at all: bad tracing

If you need some examples to get a better feeling, you can use the ones below, however there is some ambiguity within the task since we don't want to impose any strict rules but allow for a natural variety of what people find a good derendering (it is designed to be subjective). |
| Is Real | Could this digital ink have been produced by a human?

• Yes
• No

In this question go with your intuition, do you think the ink could have been traced by a [reasonable] human? |

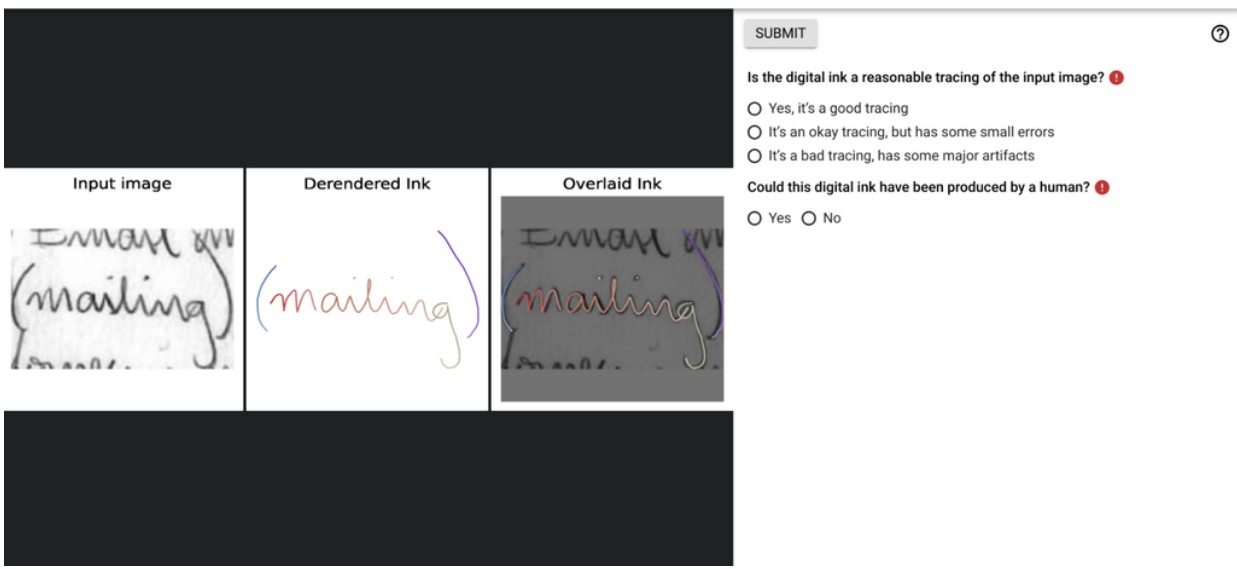

Figure 49: A screenshot of the human evaluation interface.

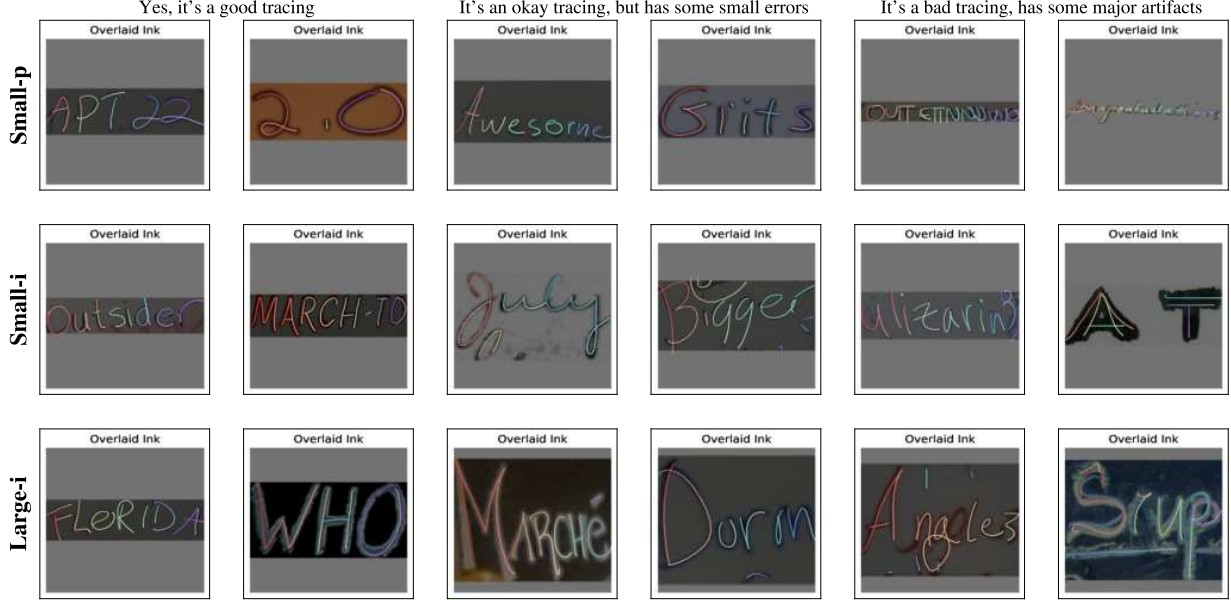

Figure 50: Examples of samples produced by our models and ratings given to them by our evaluators.

# M  Model Card

We present the model card (Mitchell et al., 2019) of our public release model **Small-p** in Table 11.

Table 11: Model card of publicly available **Small-p**.

| Model Summary | |
|---|---|
| Model Architecture | A multimodal sequence-to-sequence Transformer (Vaswani et al., 2017) model with the mT5 (Xue et al., 2021) encoder-decoder architecture. It takes text tokens and ViT (Dosovitskiy et al., 2021) dense image embeddings as inputs to an encoder and autoregressively predicts discrete text and ink tokens with a decoder. |
| Input(s) | A pair of image and text. |
| Output(s) | Generated digital ink. |
| **Usage** | |
| Application | The model is for research prototype, and the public version Small-p is planned to be released and available for the public. |
| Known Caveats | None. |
| **System Type** | |
| System Description | This is a standalone model. |
| Upstream Dependencies | None. |
| Downstream Dependencies | None. |
| **Implementation Frameworks** | |
| Hardware & Software | Hardware: TPU v5e. |
| | Software: T5X (Roberts et al., 2023), JAX/Flax (Bradbury et al., 2018), Flaxformer (Heek et al., 2023) |
| | For implementation details please refer to Appendix G. |
| Compute Requirements | Please refer to Appendix G. |
| **Data Overview** | |
| Training Datasets | The ViT encoder (Dosovitskiy et al., 2021) is pretrained on ImageNet-21k (Deng et al., 2009), mT5 encoder and decoder are initialized from scratch. The entire model is trained on the mixture of publicly available datasets described in Appendix C. |
| **Evaluation Results** | |
| Evaluation Methods | Human evaluation (reported in Section 4.5.1) and automated evaluations (reported in Section 4.5.2). |
| **Model Usage & Limitations** | |
| Sensitive Use | The model is capable of converting images to digital inks. This model should not be used for any of the privacy intruding use cases, e.g. forging handwritings. |
| Known Limitations | Reported in Section 4.7. |
| Ethical Considerations & Potential Societal Consequences | Reported in Appendix O. |

## N    Additional Visualizations

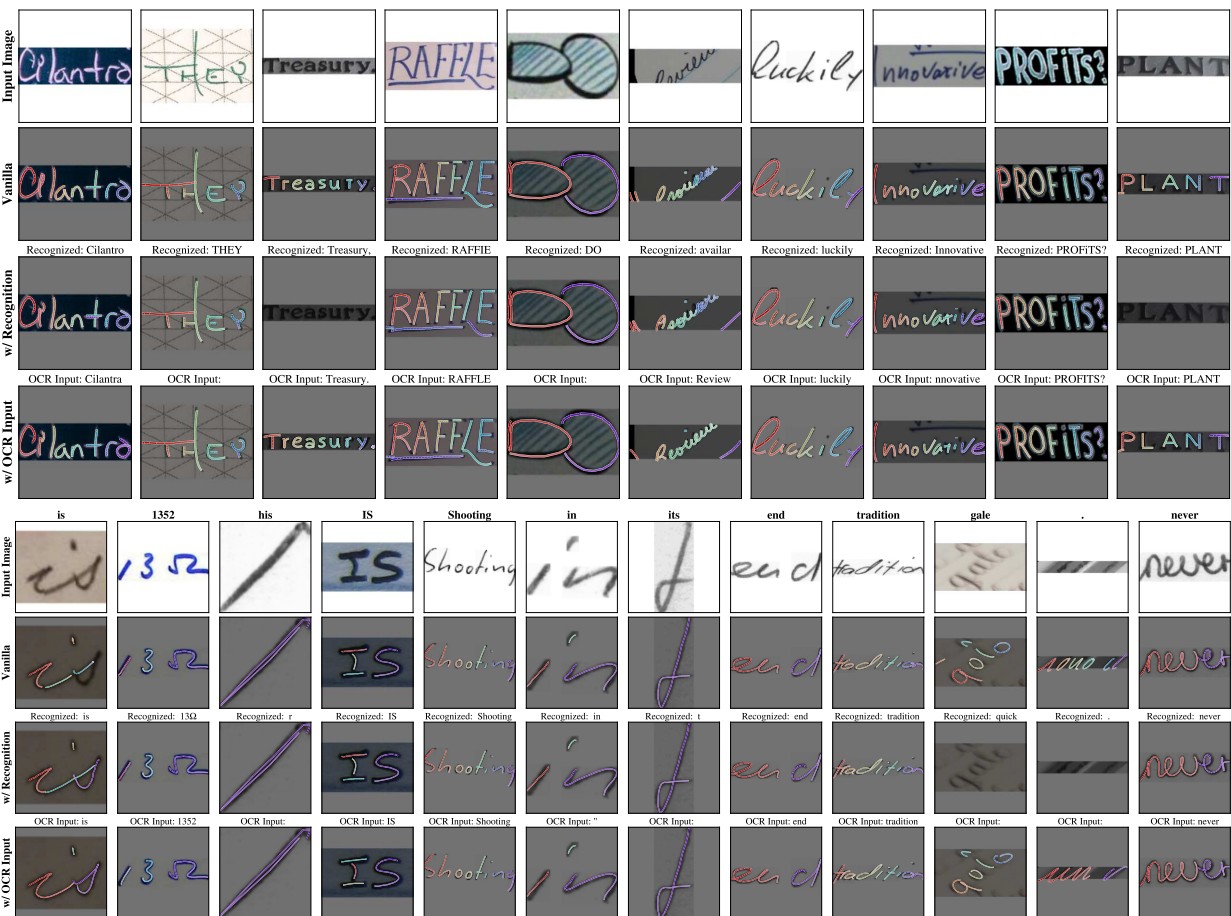

Figure 51: Results of **Small-p** on randomly selected samples from 3 public evaluation datasets.

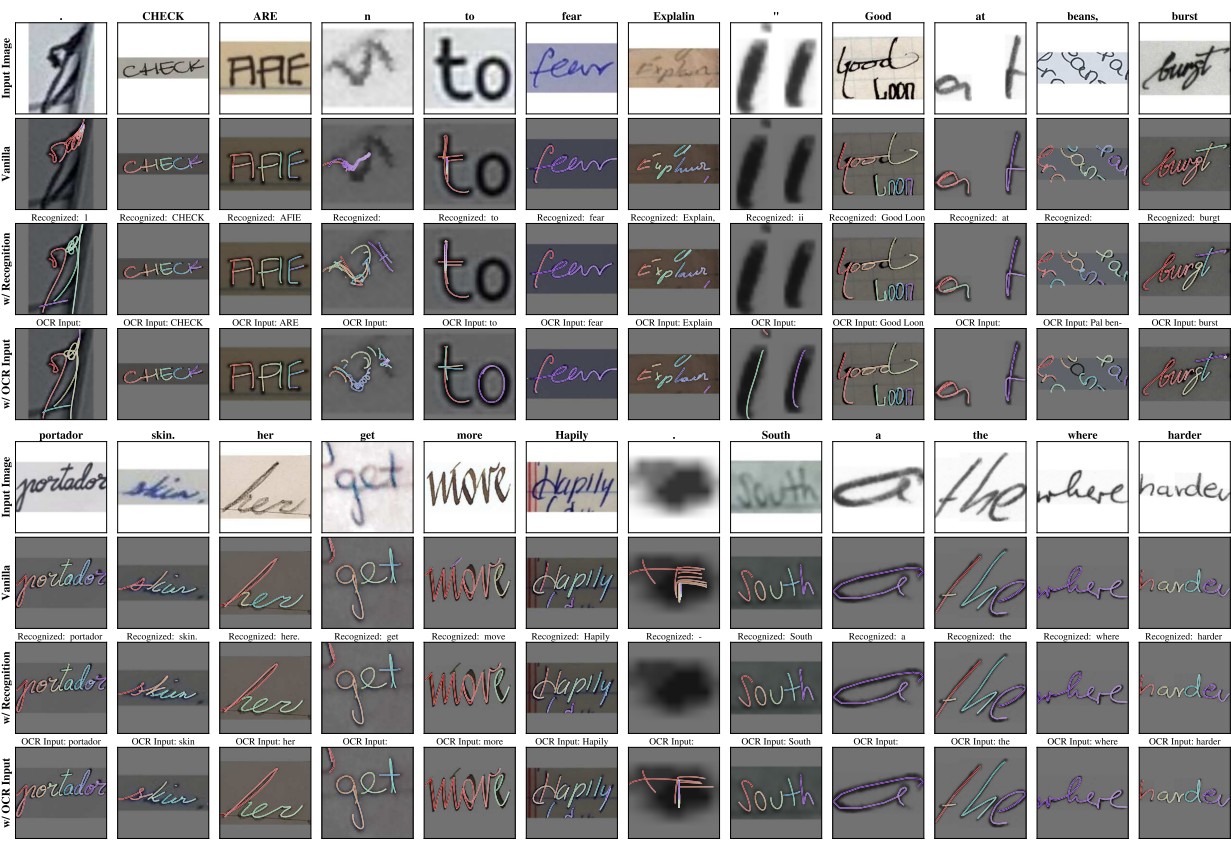

Figure 52: Results of **Small-i** on randomly selected samples from 3 public evaluation datasets.

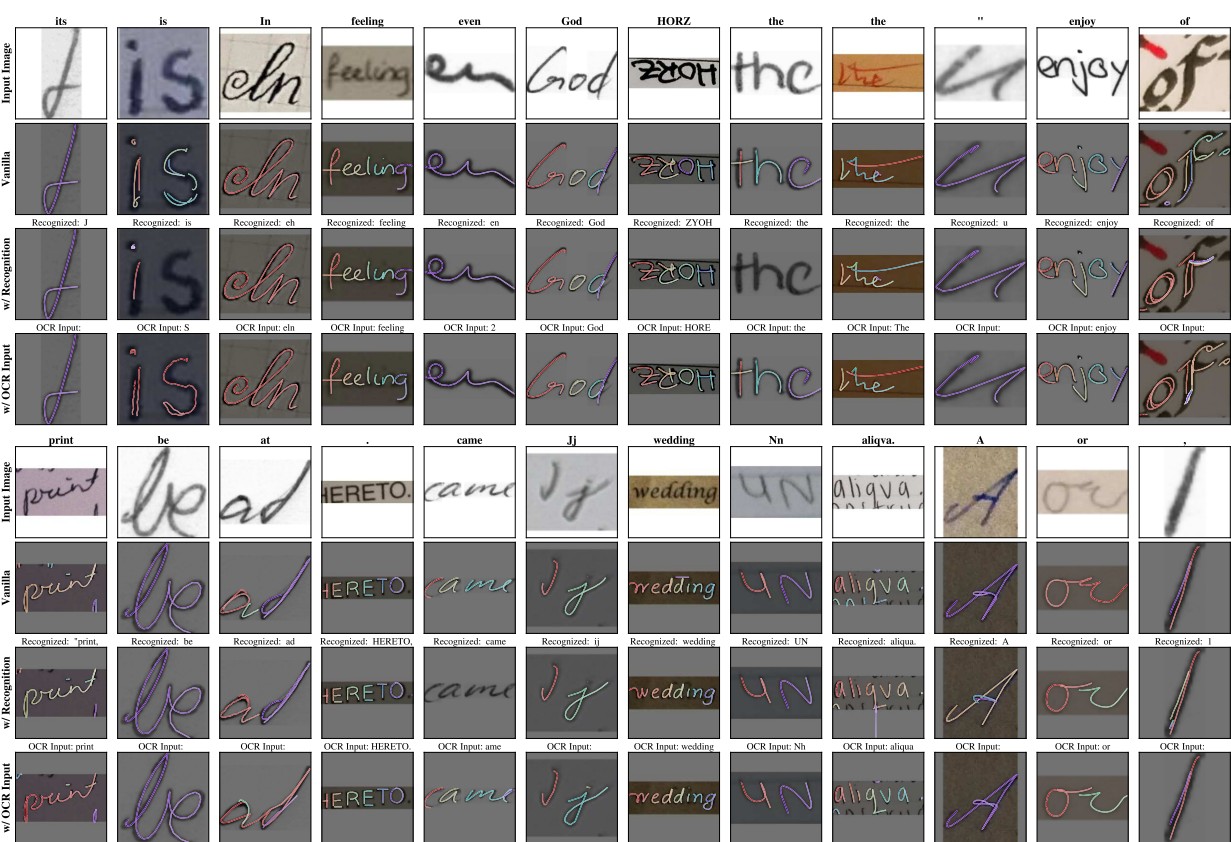

Figure 53: Results of **Large-i** on randomly selected samples from 3 public evaluation datasets.

### N.1 More Visualizations on Public Evaluation Datasets

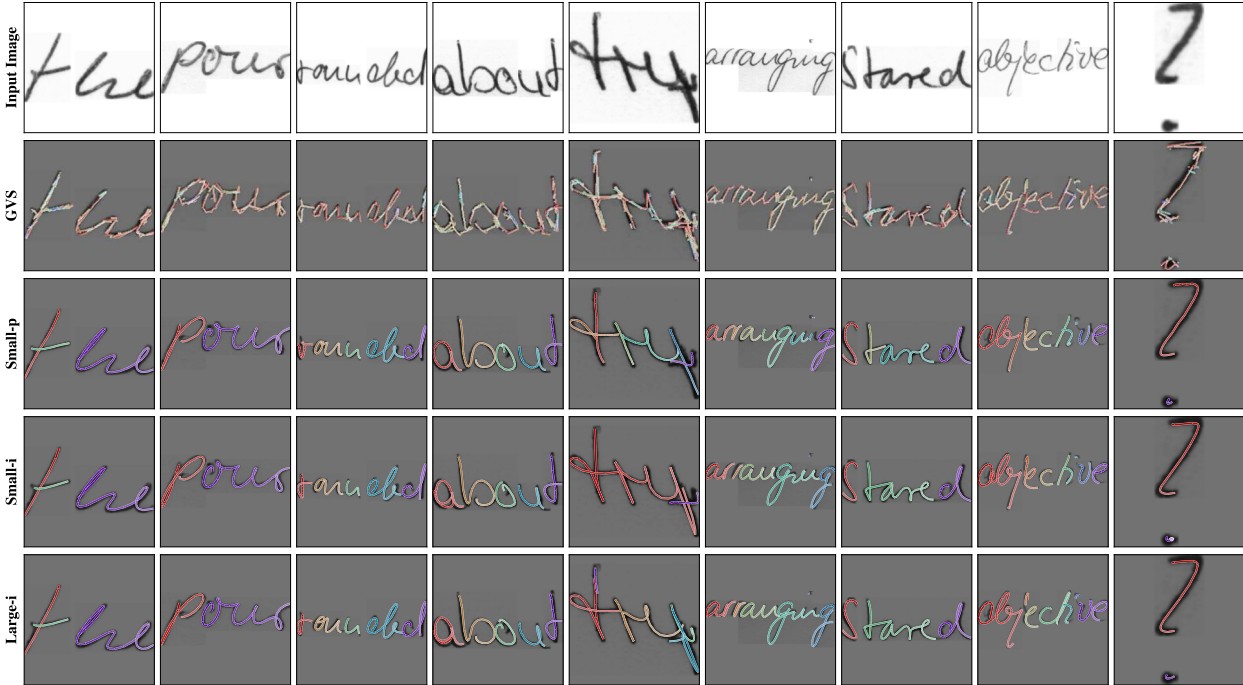

Figure 54: Comparison between **Small-i**, **Small-p**, and **Large-i** on IAM Dataset.

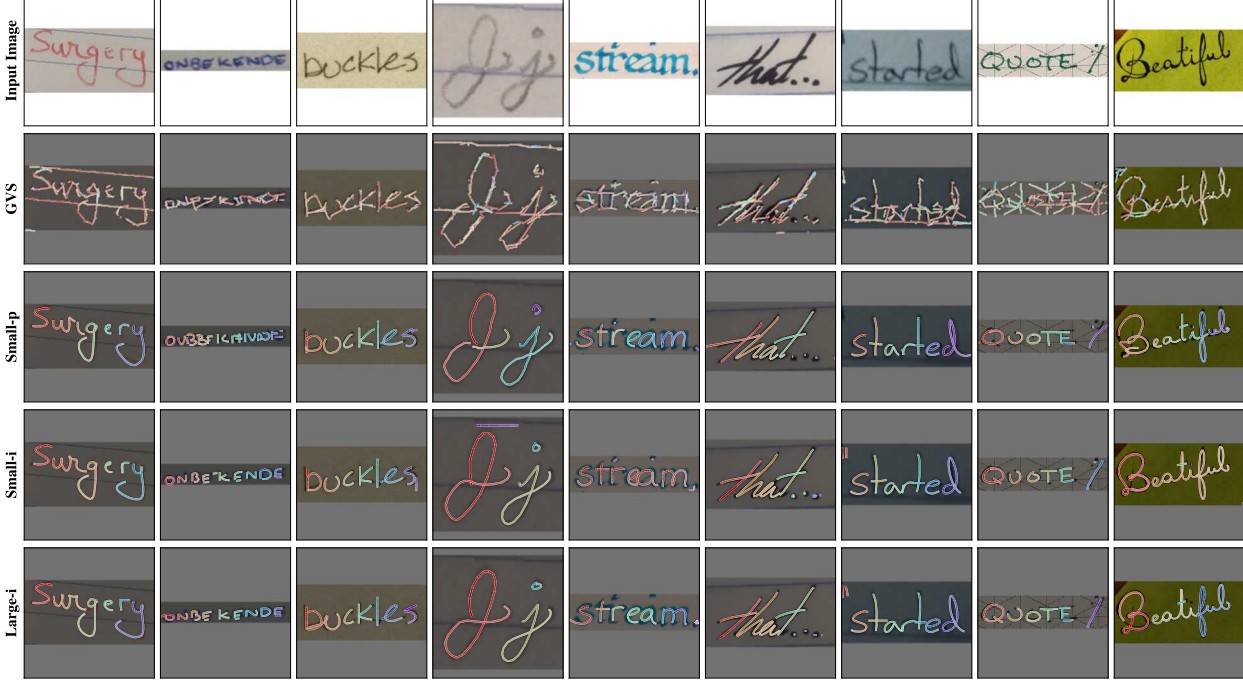

Figure 55: Comparison between **Small-i**, **Small-p**, and **Large-i** on IMGUR5K Dataset.

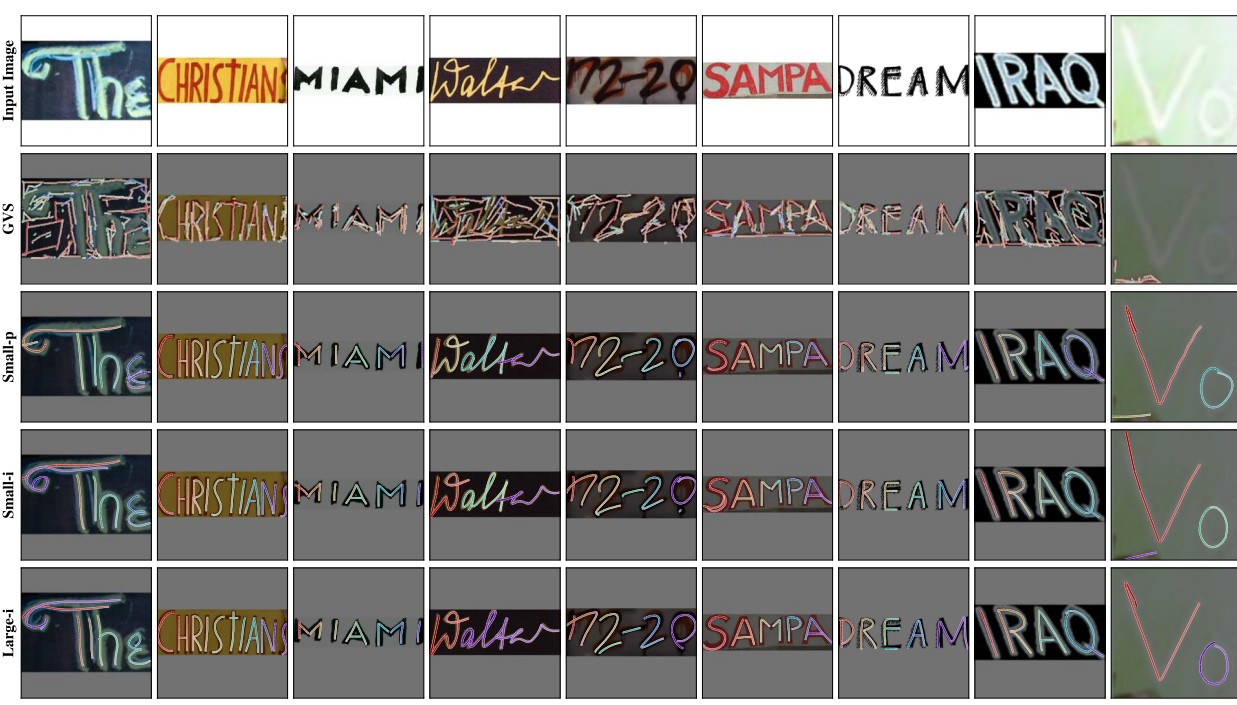

Figure 56: Comparison between **Small-i**, **Small-p**, and **Large-i** on HierText Dataset.

## O   Impact Statements

We acknowledge the importance of considering the ethical implications of InkSight. This section outlines potential concerns and benefits to promote responsible use.

### O.1   Ethical Considerations

**Forgery**   A key concern is forgery. However, InkSight is designed for offline-to-online handwriting conversion, not open-ended generation. We verify that proposed models are unable to perform open-ended ink generation. We provide random texts in task prompts of task *Derender with Text* (in Figure 3) with input images either empty or containing handwritings that do not match the label. We observe that the generated inks do not match the labels for all 100 cases inspected, signifying that the model is unable to generate the prompted text that is not present in the input image. More details in Appendix O.3.

**Bias**   Handwriting varies significantly across demographics, geographic regions, languages/scripts, age groups, and individuals with disabilities. Models trained primarily on data from dominant groups may exhibit bias, performing poorly for underrepresented styles or scripts. Ongoing efforts should focus on curating more diverse and representative training datasets encompassing a wider range of scripts, writing styles, and demographic groups. Evaluating model performance across different subgroups is crucial to identify and mitigate biases. Designing models with fairness metrics in mind is an important direction for future research.

### O.2   Positive Societal Impacts

Despite the risks, InkSight offers significant potential benefits:

**Document Digitization**   Facilitates the preservation and indexing of personal notes, historical documents, and cultural heritage artifacts beyond their textual form.

**Support for Low-Resource Languages**   By enabling the conversion of existing handwritten documents in low-resource languages into digital ink, InkSight can help generate valuable training data for online handwriting recognition systems, improving digital tool availability for these communities.

**Educational Tools**   Digital ink conversion can potentially enable new interactive learning and teaching experiences for students and teachers.

**Unsupervised Analysis via Geometric Pseudo-Labels**   Beyond supervised training data generation, InkSight's output offers a unique advantage for unsupervised analysis, particularly in low-resource or challenging scenarios. The generated digital ink sequence itself, even when semantic recognition is imperfect (e.g., for unseen languages or styles, as illustrated by the Korean example in Figure 40)captures valuable geometric and dynamic properties of the handwriting (shape, curvature, stroke connections). This ink sequence can serve as a powerful geometric pseudo-label. Offline handwriting images can then be clustered or compared based on the similarity of these ink pseudo-labels (e.g., using sequence comparison methods like DTW), leveraging purely structural information. This provides a language-agnostic alternative to OCR-based analysis, which relies on accurate text recognition and language models, allowing for grouping or analysis based on visual writing patterns even when the text content is unknown or poorly recognized.

### O.3   Open-ended Generation

As illustrated in Figure 57, our experiments with **Small-p**, **Small-i**, and **Large-i** models demonstrate that when prompted with text not present in the original image, the models fail to generate accurate corresponding handwritten strokes. We regard this behavior as a natural safeguard against potential misuse, such as forgery or unauthorized reproduction of handwriting styles. Since the models can only reconstruct handwriting patterns from the input image rather than generating arbitrary new content, they inherently restrict attempts to synthesize fraudulent handwritten content.

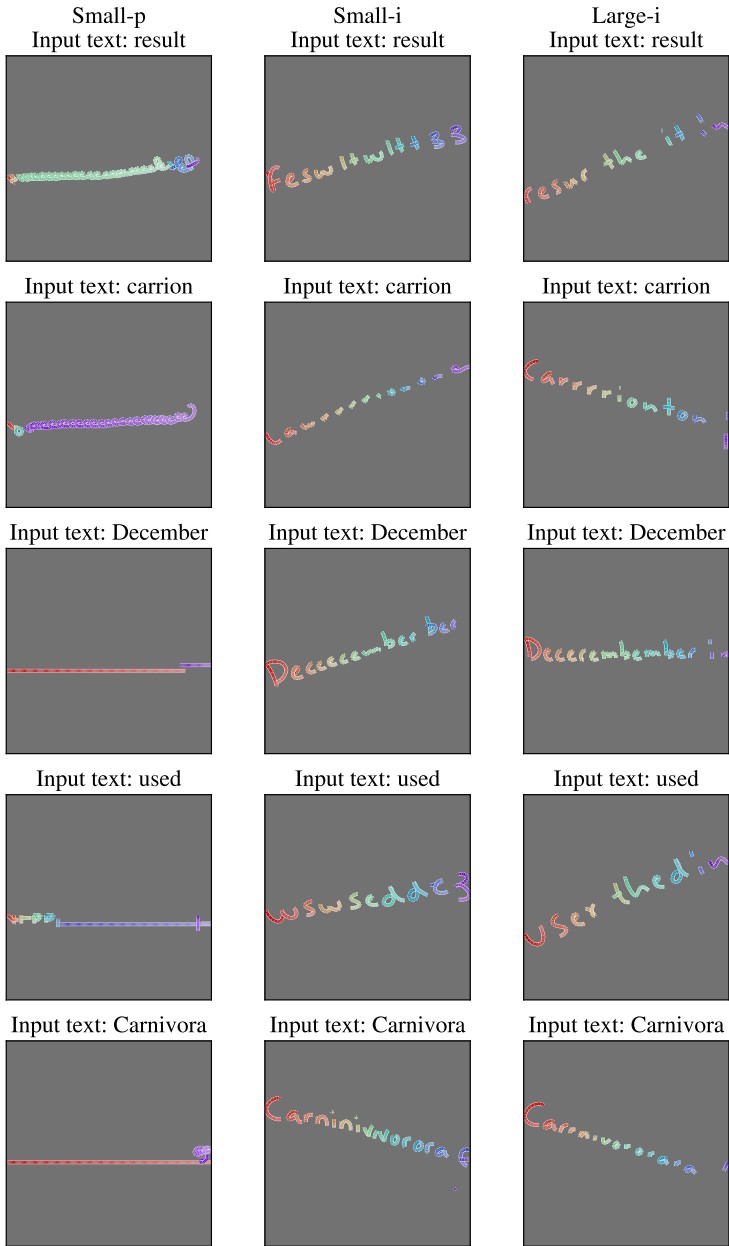

Figure 57: **Inability to perform open-ended generation.** Prompting **Small-p**, **Small-i** and **Large-i** with text that is not presented in the image does not produce digital inks that match the textual input.

