# OpenReview forum: "InkSight: Offline-to-Online Handwriting Conversion by Teaching Vision-Language Models to Read and Write"
_TMLR — Accepted by TMLR_

### Review · Reviewer_NKQU · 2025-02-11

**Summary Of Contributions:**

This paper explores a method for converting handwritten text in images into digital ink, which represents the pen trajectory for multiple characters.
To accomplish this, this paper proposes a model called InkSight, which consists of a ViT encoder and an mT5 encoder-decoder.

**Audience:**

Yes

**Broader Impact Concerns:**

I have not ethical implications for this paper.

**Claims And Evidence:**

Yes

**Requested Changes:**

## Major Concerns
### weakness 1
This paper should discuss specific applications that require digital ink rather than just pen trajectory for individual characters.
It would be beneficial to demonstrate a concrete application that necessitates digital ink.

### weakness 2
I recommend creating a small dataset that includes ground truth digital ink and using it to evaluate the proposed method.

## Minor Concerns
### weakness 3
I recommend evaluating the proposed method on the datasets for pen trajectory recovery of individual characters.

**Strengths And Weaknesses:**

## Strength
1. Extending pen trajectory recovery from individual characters to text in wild images is a non-trivial challenge.

2. The qualitative results are impressive.
The proposed model appears to successfully convert handwritten text into digital ink.

3. The training scheme, which utilizes both synthetic and real data, appears to be plausible.

## Weakness
1. I am unsure about the practical applications of converting handwritten text into digital ink.
Previous studies on pen trajectory recovery have primarily focused on applications such as signature verification and calligraphy synthesis.
How important is the extension from individual characters to full text in pen trajectory?

2. The evaluation dataset does not appear to contain ground truth digital ink.
As a result, the quantitative evaluation relies on the recognition results of the traced pen trajectory.
It is unclear whether the quantitative evaluation truly correlates with the quality of the digital ink.

3. The baseline is weak, as the method is not specifically designed for pen trajectory.
The suitability of the proposed language model-based approach for this domain remains unclear.

---

> ### Author Response · Authors · 2025-02-18
>
> **We thank Reviewer NKQU for their insightful feedback. We have carefully considered their comments and addressed each point below and improved the manuscript.**
>
> ## Practical Applications (Major Concern)
> We thank the reviewer for prompting us to further clarify the practical applications of full-text digital ink conversion. While Section 1 (Introduction) already touches upon these applications, and Section 4.5.2 demonstrates the use of derendered ink for improving handwriting recognition, we agree that a more explicit and detailed discussion is beneficial. To address this, we have:
> 1. **Strengthened the motivation in the main Introduction (Section 1):** We now more clearly contrast our image-to-ink conversion approach with OCR and emphasize the unique advantages of preserving the process of writing.
> 2. **Separated and emphasized the paragraph highlighting the interest in both academia and industry in Section 1 (Introduction):** We have ensured this paragraph is prominently positioned to underscore the real-world relevance and broad appeal of **image-to-ink conversion**.
>
>
> We believe these changes enhance the perceived significance of our work. The key practical applications we now explicitly highlight include:
> 1. **Enhanced Note-Taking and Archiving:** **Image-to-ink conversion** directly addresses the needs of users who prefer pen-and-paper note-taking but desire the benefits of digital notes. As we mention in the Introduction, similar **image-to-ink conversion** technology is already present and valued in applications like **Notability (Not, 2023)**. InkSight provides a robust and generalizable approach to this valuable functionality.
> 2. **Improved Accessibility:** **Image-to-ink conversion** offers new avenues for creating accessible handwritten documents, going beyond simple text extraction to capture structural and stylistic information for assistive technologies.
> 3. **Synthetic Data Generation for Handwriting Recognition:** Critically, as demonstrated by our quantitative results in Section 4.5.2 and Appendix I, **image-to-ink conversion** enables the generation of high-quality synthetic training data for online handwriting recognition models. This is particularly valuable for improving recognition accuracy and expanding language support, especially for low-resource languages in the digital ink domain.

---

> > ### Author Response · Authors · 2025-02-18
> >
> > ## Evaluation Dataset (Major Concern)
> >
> > We appreciate the reviewer's careful consideration of our evaluation methodology. We would like to clarify that, while the majority of our evaluation uses OCR datasets due to the scarcity of large-scale paired image/ink data (as discussed in Section 2 and Section 4.5), we do include a smaller dataset with ground truth digital ink. This "golden" set, described in Section 4.5.1 and Appendix K, consists of human-traced samples from the HierText dataset. This dataset, though smaller in scale, allows us to:
> >
> > 1. **Conduct a human evaluation (Section 4.5.1):** This evaluation directly assesses the perceptual and semantic quality of the generated ink compared to expert-produced ink, providing a crucial qualitative assessment.
> > 2. **Establish a reference point for automated metrics (Section 4.5.2):** We use the expert-traced data to determine a baseline performance level for our automated metrics, allowing for a more meaningful interpretation of the results on the larger OCR datasets.
> >
> > Furthermore, we open-source a subset of our expert-traced dataset to support reproducibility and future research on this topic **(Section 1, Contributions)**. While we acknowledge that a larger, more diverse dataset with ground truth ink traces would be ideal, this could be addressed in future work building upon our research.

---

> > > ### Author Response · Authors · 2025-02-18
> > >
> > > ### Evaluation on Single Character (Minor Concern)
> > >
> > > We appreciate the reviewer's suggestion regarding evaluation on single-character datasets. We would like to clarify that our framework is primarily designed for full-page derendering, mirroring real-world scenarios. Our chosen evaluation benchmarks, as described in Section 4.1 and Section 4.5, reflect this focus, including datasets like IAM, IMGUR5K, and HierText, which contain words and lines of text. It's important to note, however, that these datasets **do incidentally include instances of single, isolated characters**, particularly in datasets like IAM. Therefore, our evaluation already encompasses some single-character cases, even if they are not the primary emphasis. Examples of this can be seen in our supplementary jupyter notebook. Our reported results, therefore, provide some indication of performance on single characters within a broader context.
> > >
> > > We agree that a dedicated, focused evaluation on exclusively single-character datasets, using metrics specifically designed for that task, could be valuable, *particularly for certain languages or applications where isolated character recognition is paramount*. However, as detailed in our previous response, adapting our model and evaluation pipeline to these specialized datasets would represent a substantial shift in focus and is beyond the scope of the current paper. We welcome the use of InkSight as a baseline for comparison in such evaluations. Our publicly released model and code provide a readily available starting point.

---

> > > > ### Author Response · Authors · 2025-02-18
> > > >
> > > > ### Weak Baseline (Minor Concern)
> > > >
> > > > We thank the reviewer for pointing out the limitations of the GVS baseline. We agree that GVS is not specifically designed for pen trajectory recovery. However, as detailed in **Section 4.3** and **Appendix D (where we list all related approaches and their limitations)**, there is a scarcity of publicly available code, training data, and pre-trained models for this specific task. GVS, while designed for sketch vectorization, represents the closest available and reproducible approach that addresses a similar problem (vectorizing line drawings). We chose it to provide some point of comparison, acknowledging its limitations.
> > > >
> > > > We believe our vision-language model-based approach offers several advantages:
> > > >
> > > > - Learned Priors: As discussed in **Section 1** and **Section 3**, leveraging vision-language models allows us to incorporate learned reading and writing priors, which are crucial for handling the complexities of real-world handwritten text (occlusions, varying styles, etc.). This is a key distinction from purely geometric approaches.
> > > > - Scalability and Reproducibility: Our approach, based on standard Transformer components, is inherently scalable to longer sequences and higher resolutions, as discussed in **Section 3**. As detailed in **Section 3.1**, **Appendix G** and **Appendix L**, our approach is constructed with publicly available building blocks, with detailed implementation and a model card for Small-p, fostering the reproducibility.
> > > >
> > > > We are confident that this detailed explanation effectively addresses the reviewer's concern regarding the baseline and highlights the key advantages of InkSight.

---

> > > > > ### Comment · Reviewer_NKQU · 2025-04-02
> > > > > **Response about minor concern**
> > > > >
> > > > > While I am not entirely sure to what extent the proposed model is appropriate for the task involving digital ink, I acknowledge that the human study supports the advantages of the proposed model over the baselines.

---

> > ### Comment · Reviewer_NKQU · 2025-04-02
> > **Response about the major concern 1**
> >
> > Thank you for addressing to support motivation, but I am still not sure how digital ink is essential.
> > I would appreciate it if the authors could not only provide references but also present concrete applications where digital ink is required.
> > For example, how is digital ink utilized in Notability?

---

> > > ### Comment · Reviewer_NKQU · 2025-04-02
> > > **Response about the major concern 2**
> > >
> > > Thank you for the clarification. I now understand that the golden set from the HierText dataset corresponds to the ground truth.

---

> > > ### Author Response · Authors · 2025-04-03
> > >
> > > Thank you for your time to read our response and for highlighting the need for more concrete examples regarding the essential value of digital ink.
> > >
> > > We agree and will revise Section 1 and Appendix to detail specific functionalities enabled only by digital ink's stroke-level, sequential data (e.g., replay, fine-grained editing, flexible formatting, reusing, etc.)

---

> > > > ### Comment · Reviewer_NKQU · 2025-04-04
> > > > **Response about the major concern 1**
> > > >
> > > > I appreciate the authors taking the time to address the concern.
> > > > The central issue I see in this paper is whether the proposed task has a meaningful impact.
> > > > I hope the authors provide a thorough discussion and clear demonstration of how these applications are beneficial.

---

> > > > > ### Author Response · Authors · 2025-04-04
> > > > > **Response to major concern 1**
> > > > >
> > > > > We thank the reviewer for their feedback and for raising the point about clearly demonstrating the meaningful impact and benefits of our proposed task: offline-to-online handwriting conversion (derendering). We here provide a thorough discussion and clear demonstration, although we believe the paper already provides substantial evidence and justification, we are happy to reiterate and clarify given the length of the paper.
> > > > >
> > > > > The core benefits and impactful applications enabled by converting offline handwriting to digital ink (which preserves stroke order and geometry, unlike OCR) are discussed primarily in Section 1, with supporting evidence from other sections:
> > > > >
> > > > > 1. **Core benefits and interests in research and industry (Section 1, Section 2):** Derendering by bridging the physical-digital gap, allows users who prefer pen-and-paper to gain digital benefits like **fine-grained stroke-level editing/erasure**, reformatting, and integration with other digital content while preserving the unique personal touch. This significantly improves workflow efficiency and information management. The interest has been proven by the lines of research on pen trajectory recovery and line drawing vectorization, and we reviewed the line of research in Section 2 and found they were brittle for derendering any input conditions. Furthermore, in the industry, the existence of features based on this principle in popular apps like Notability (referenced in Section 2) underscores the real-world demand and impact for millions of users. For information on notability, we provide: reference to notability site [1]. We also conducted a comparison with Notability, and have attached one anonymized slide for illustration: [2] another example: [3].
> > > > > 2. **Improved Online Handwriting Recognition (Section 4.5.2 Figure 9, Appendix I):** Our experiments quantitatively show that using the digital ink derendered by InkSight from offline images improves the performance of online Handwriting Recognition (HWR) models. This provides a crucial pathway to generate valuable training data for online HWR in low-resource languages, where digital ink data is often scarce but offline documents may exist. This directly addresses a significant bottleneck in expanding digital tool accessibility.
> > > > > 3. **Preservation and Analysis (Appendix N):** Generating digital ink facilitates the preservation of historical documents and enables stylistic and forensic analysis by capturing stroke dynamics (order, geometry) lost in simple digitization or OCR. This capability is also shown in the Alice's Adventures in Wonderland example in the paper’s first figure (Figure 1).
> > > > >
> > > > > We hope these points, discussed and evidenced in the referenced sections, demonstrate the significant and meaningful impact of offline-to-online handwriting conversion. The ability to bridge the physical-digital divide, while preserving the unique characteristics of handwriting unlocks capabilities that are highly sought after by both academia and industry interest.
> > > > >
> > > > > We have ensured that Section 1 in particular, now more explicitly details how digital ink enables these specific functionalities and benefits. We hope this clarification addresses the reviewer's concern about the task's impact.
> > > > >
> > > > > [1]: https://support.gingerlabs.com/hc/en-us/articles/5044440428570-Image-to-Ink-Conversion
> > > > > [2]: https://ibb.co/Q3X8CTZ6
> > > > > [3]: https://ibb.co/7JBcLQT9

---

### Review · Reviewer_iQTZ · 2025-03-12

**Summary Of Contributions:**

This paper introduces a system capable of transforming arbitrary photos of handwritten notes into digital ink (derendering), enabling conversion from offline handwriting to online handwriting without specialized equipment. Unlike prior methods limited by geometric properties, this approach utilizes learned reading and writing priors, enhancing precision in stroke order and dynamics. Evaluations on the HierText dataset show the model’s effectiveness, with 87% of outputs rated as accurate tracings and 67% indistinguishable from human pen trajectories. Additionally, the method demonstrates robustness beyond its training domain, generalizing to various handwriting styles, sketches, and complex scenarios. The authors publicly release model weights and inference code.

**Audience:**

No

**Claims And Evidence:**

Yes

**Requested Changes:**

My main requested changes would be: can you address the above mentioned limitations
- thoroughly discussing the inference speed given different image condition, using FLOPS or Token/s. Also, how to improve the overall system performance if handle a page full of notes. Even we need multiple times of model inference, we may can still optimize the whole system performance.
- how to improve the system performance in handling very wider stroke width.

**Strengths And Weaknesses:**

As firstly, I'd like to kindly point out some weakness in my mind:
- I did not see particularly technical contributions to address the mentioned two challenges: Limited Supervised Data and Scalability to Large Images.
    - The paper indeed proposed some data augmentation strategies in Section 3.2. However, even the data augmentation includes changing of stroke width, but as mentioned in Appendix J (page 36), "model’s performance deteriorates when processing samples with wider stroke widths or significant variations in stroke width". This shows the proposed strategy is not that successful even the numbers in ablation studies table 3 are good.
    - As mentioned in Section 4.7 "Limitations", "the proposed method can derender only several characters or words in one model inference call", while this is exactly the second challenge mentioned in the beginning.

- The whole paper gives me the feeling that some domain-specific data is being collected and the VLM is being fine-tuned. I am unsure if any of the studied components can benefit other domains.

Besides the above fundamental weaknesses, this paper studied the particular question very deep. Figure 4 is intriguing, and overall the paper is easy to follow. Also, they included failure cases analysis which are my particularly interested.

In general, I believe this paper can serve as a great reference for future research in this particular direction.

---

> ### Author Response · Authors · 2025-03-23
>
> We sincerely thank the reviewer for their thorough review of the paper and appendix and for the valuable feedback provided. We appreciate the time and effort they invested in understanding our work in detail. In response to the reviewer’s comments, we have updated the manuscript to address the points raised. These changes aim to improve clarity and better reflect the contributions of our work.

---

> ### Author Response · Authors · 2025-03-23
>
> ## Limited Supervised Data & Data Augmentation
>
> We acknowledge the reviewer's comment that our data augmentation strategy might be perceived as "not that successful," even though the ablation studies in Table 3 show positive results. We believe there might be a slight misunderstanding of the purpose and effectiveness of our data augmentation.
>
> We present an ablation study in Table 3, which specifically demonstrates the results of adding the proposed multi-task training and data augmentation. Given the significant contribution of data augmentation, as evidenced by the marked performance decline when data augmentation is removed, it is clear that this strategy is important in addressing the challenge of Limited Supervised Data.
> We acknowledge the limitations discussed in Appendix J and Section 4.7 regarding extreme stroke widths. However, we view these as defining the current boundaries of our model rather than contradicting the overall effectiveness of data augmentation. Along with the reviewer’s suggestions, there may be certain ways to  enhance robustness, especially for diverse and extreme stylistic variations, we see these as valuable directions for future work, as they extend beyond the primary focus of this paper: demonstrating the core effectiveness of a vision-language model for arbitrary image-to-digital ink conversion using multi-task learning and a data augmentation strategy to learn reading and writing priors.

---

> ### Author Response · Authors · 2025-03-23
>
> ## Scalability to Large Images
>
> We thank the reviewer for raising the point about scalability and the limitation mentioned in Section 4.7 regarding the number of characters/words processed in a single model inference call. We believe there may be a slight misunderstanding regarding the overall system's ability to handle full pages, and we appreciate the opportunity to clarify this.
>
> It is correct that, due to the fixed context window length of the Transformer-based architecture, a single model inference call is limited to processing a few words or characters. This is a limitation of the individual model inference, not the overall system for full-page derendering.
>
> As illustrated in Figure 2 and described in Section 3, we have developed a complete pipeline that enables full-page derendering by leveraging:
>
> 1. Pre-segmentation: We use a word-segmentation model (e.g., our own, or alternatives like docTR, tesseract) to pre-segment the full page into individual word bounding boxes. This is a standard and efficient preprocessing step.
>
> 2. Parallel Word-Level Inferences: We then perform parallel model inferences on each of these word-level segments. This allows us to effectively process the entire page, overcoming the single-call context window limitation.
>
> 3. Post-processing and Stitching: Finally, we stitch the individual word-level digital ink outputs back together to reconstruct the full-page derendered result.
>
> Therefore, while the model itself has a limited context window, we build efficient system design, resampling, and sequence reduction to perform full-page derendering. We demonstrate full-page derendering results throughout the paper, most notably in Appendix B and supplementary materials, which showcases multiple examples of complete handwritten pages successfully converted to digital ink.
>
> We have clarified this important distinction between model-level limitations (and the steps taken to address them) and system-level capabilities in Section 4.7 (Limitations) to avoid any further misunderstanding.

---

> > ### Author Response · Authors · 2025-03-23
> >
> > ## Generalizability
> >
> > We thank the reviewer for raising the important question of generalizability beyond handwritten text. While our current evaluation is specific to handwriting derendering, we emphasize two key points: 1) the demonstrated practical value and demand for this specific system, and 2) the potential for broader applicability of the InkSight framework.
> >
> > First, handwritten text derendering (image-to-digital ink conversion) addresses a significant real-world need as evident from consumer application solutions to this problem mentioned in Section 1.
> > Second, our core technical contribution extends beyond fine-tuning a VLM. We propose a novel framework for image-to-vector conversion, leveraging the learned priors of vision-language models. This framework, which integrates a ViT encoder, an mT5 sequence-to-sequence model, and a multi-task training strategy, is not intrinsically limited to handwriting. The fundamental principle – transforming raster images of line drawings into structured vector representations – is potentially applicable to other domains.
> >
> > While demonstrating broader applicability is beyond the scope of this paper (which focuses on establishing feasibility and effectiveness for the challenging task of handwriting derendering), we believe exploring applications like sketch, diagram, or mathematical formula vectorization is a valuable direction for future work, as noted in Section 5.

---

> > > ### Author Response · Authors · 2025-03-23
> > >
> > > ## Inference Profiling
> > >
> > > We thank the reviewer for their questions about inference speed and full-page optimization. We've added details to Appendix G.
> > >
> > > As the reviewer correctly notes, our system processes images through a series of word-level inferences. Crucially, **however, our system design does not require handling different image conditions separately** as described in the full-page derendering section. All input images, regardless of their specific visual characteristics are processed through the same standardized pipeline and model. This allows for batch processing, improving throughput. We use a consistent batch size 32 for fairness. Appendix G reports inference speed in Tokens/s for publicly available Small-p on a range of hardwares.

---

### Review · Reviewer_2tDr · 2025-03-20

**Summary Of Contributions:**

The paper introduces InkSight, a novel and robust method for converting images of handwritten notes into digital ink by leveraging vision-language models, specifically mT5 and a pretrained ViT. To the best of my knowledge, this is the first work addressing offline-to-online derendering. Extensive experiments validate the effectiveness of the proposed system, demonstrating that it successfully reconstructs online handwritten text that is both semantically and geometrically aligned with the input images. This derendering process proves valuable for generating additional training data to enhance handwriting recognition. Moreover, the approach generalizes well to unseen data, handling diverse visual conditions beyond the training distribution.

**Audience:**

Yes

**Broader Impact Concerns:**

A section on ethical implications and a Broader Impact Statement is essential, given the work’s focus on handwritten text processing. Privacy concerns arise as handwritten documents may contain personally identifiable information, requiring proper data anonymization and compliance with regulations. Bias in handwriting recognition must also be discussed, e.g., across diverse scripts, demographics, and writing styles, including for individuals with disabilities. The system’s potential misuse in forgery, document tampering, or identity theft necessitates safeguards. At the same time, the technology can benefit document digitization and accessibility, particularly for low-resource languages and historical documents. Responsible deployment should include clear licensing. Ensuring open-source accessibility while preventing misuse is critical. Adding these discussions will strengthen the paper’s ethical considerations and promote the responsible adoption of the proposed approach.

**Claims And Evidence:**

Yes

**Requested Changes:**

+ Consider adding experiments to evaluate whether the proposed derendering approach improves offline handwriting recognition. Since online handwriting recognition is generally easier, demonstrating its impact on offline recognition would further highlight the practical utility of the method.

+ Historical degraded documents such as H-DIBCO can also be considered for evaluation.

+ Include a detailed analysis of the model’s generalisation to unknown writers, diverse writing styles, and new languages or scripts. Providing both quantitative metrics and qualitative examples would strengthen the paper’s claims of robustness.

+ Expand the discussion on failure cases by offering deeper insights into the model’s limitations. Analysing where and why the approach struggles could help understand its boundaries and potential areas for improvement.

+ Add a section on ethical implications and broader impact. (Refer Broader Impact Concerns)

**Strengths And Weaknesses:**

**Strength:**
+ **Novel Task and Effective Solution:** The proposed task is novel and holds significant potential for preserving handwritten notes and historical documents while enhancing the machine readability of handwritten text. Although the approach involves a straightforward architectural combination of mT5 and a pretrained ViT (similar to PaLI), it proves to be highly effective, demonstrating exceptional performance in derendering.

+ **Comprehensive Experimental Analysis:** The experiments are comprehensive and strongly support the claims made in the paper. Ablation studies and visual results further validate the design choices, emphasizing the importance of the Derendering with Text task during inference for achieving optimal results. Qualitative analysis highlights that the small variant (small-i) achieves notable performance gains over the baseline GVS, while the best-performing variant (Large-i) exhibits greater robustness to diverse visual conditions and image styles.

+ **Generalization to out-of-domain data:** Additionally, the proposed approach generalises well to out-of-domain simple sketches. Both human and automatic evaluations (character-level F1 score, exact match accuracy) substantiate the model's performance. Furthermore, the paper experimentally demonstrates that combining derendered data with real online handwritten text enhances online handwritten recognition.

Overall, the paper is well-structured, well-presented, and a pleasure to read, and it has a lot of value for the handwritten image analysis community.


**Weakness:**
+ The paper does not provide empirical evidence on whether the proposed derendering approach improves offline handwriting recognition. Since online handwriting recognition is generally considered easier than offline recognition, experiments in this direction would strengthen the work’s impact.

+ The generalization of the proposed model to unknown writers, writing styles, and new languages or scripts is not thoroughly analyzed. A detailed quantitative and qualitative evaluation in this regard would enhance the completeness of the study.

+ While the limitations of the model are acknowledged, the paper offers limited insight into failure case analysis, making it difficult to understand where and why the approach struggles.

---

> ### Author Response · Authors · 2025-03-27
>
> We sincerely thank the reviewer for their thorough review of the paper and appendix and for the valuable feedback provided. We appreciate the time and effort they invested in understanding our work in detail. In response to the reviewer’s comments, we have updated the manuscript to address the points raised. These changes aim to improve clarity and better reflect the contributions of our work.

---

> > ### Author Response · Authors · 2025-03-27
> >
> > ## Offline vs Online handwriting improvement
> >
> > We appreciate the reviewer’s suggestion to evaluate the effect of derendering on offline recognition rather than online recognition. While we agree that online recognition is generally easier, this is only true when in-domain data is available to train models for that specific task.
> >
> > In the “Related Work” section ("Dataset availability"), we highlight that available data is scarce. Consequently, extending offline datasets to online by converting them while preserving the ground truth textual labels could significantly benefit the development of online recognition systems (also as shown in Fig. 9 and Appendix I).
> >
> > At the same time, offline optical character recognition (OCR) datasets are more abundant, covering handwritten texts across various scripts (e.g., Latin: RIMES [1], NIST [2], IMGUR5K [3], GNHK [4]; Arabic: KHATT [5]; Indic: IIIT-INDIC-HW-WORDS [6]; Chinese: SCUT-EPT [7]), as well as other domains like forms, printed documents, and street signs (e.g., FUNSD [8], HierText [9, 10], SVT [11, 12], and C-SVT [13]).
> >
> > We agree with the reviewer that in domains where labeled online data is available, but only unlabeled offline data exists, our approach could be beneficial for offline handwriting recognition. It can be helpful to generate synthetic data by derendering the offline images with recognition and then using the labels for offline recognition training. However, since we are unaware of such a domain or situation, we do not consider this to be a major contribution of the proposed methodology.
> >
> >
> > ---
> > [1] **Kim, J., Glassman, E. L., Monroy-Hernández, A., & Morris, M. R. (2015).** RIMES: Embedding interactive multimedia exercises in lecture videos. *Proceedings of the 33rd Annual ACM Conference on Human Factors in Computing Systems (CHI ’15)*, 1535–1544. ACM.
> >
> > [2] **National Institute of Standards and Technology (NIST). (n.d.).** *NIST Special Database 19: NIST Handprinted Forms and Characters Database.* Retrieved March 27, 2025, from https://www.nist.gov/srd/nist-special-database-19[3] **Krishnan, P., Kovvuri, R., Pang, G., Vassilev, B., & Hassner, T. (2021).** *TextStyleBrush: Transfer of Text Aesthetics from a Single Example.* arXiv preprint arXiv:2106.08385.[4] **Lee, A. W. C., Chung, J., & Lee, M. (2021).** GNHK: A dataset for English handwriting in the wild. *Proceedings of the International Conference on Document Analysis and Recognition (ICDAR).*
> >
> > [5] **Mahmoud, S. A., Ahmad, I., Alshayeb, M., Al-Khatib, W. G., Parvez, M. T., Fink, G. A., Märgner, V., & Abed, H. E. (2012).** *KHATT: Arabic offline handwritten text database. Proceedings of the 2012 International Conference on Frontiers in Handwriting Recognition (ICFHR 2012), 449–454. IEEE.*
> >
> > [6] ***Gongidi, S., & Jawahar, C. V. (2021).** iiit-indic-hw-words: A dataset for Indic handwritten text recognition. In J. Lladós, D. Lopresti, & S. Uchida (Eds.), Document Analysis and Recognition – ICDAR 2021 (Lecture Notes in Computer Science, Vol. 12824). Springer, Cham.*
> >
> > [7] ***Zhu, Y., Xie, Z., Jin, L., Chen, X., Huang, Y., & Zhang, M. (2018).** SCUT-EPT: A new dataset and benchmark for offline Chinese text recognition in examination paper. IEEE Access. IEEE.*
> >
> > [8] ***Jaume, G., Ekenel, H. K., & Thiran, J.-P. (2019).** FUNSD: A dataset for form understanding in noisy scanned documents. Proceedings of ICDAR-OST.*
> >
> > [9] **Long, S., Qin, S., Panteleev, D., Bissacco, A., Fujii, Y., & Raptis, M. (2022)*.** Towards end-to-end unified scene text detection and layout analysis. Proceedings of the IEEE/CVF Conference on Computer Vision and Pattern Recognition (CVPR).*
> >
> > [10] **Long, S., Qin, S., Panteleev, D., Bissacco, A., Fujii, Y., & Raptis, M. (2023).** *ICDAR 2023 competition on hierarchical text detection and recognition. arXiv preprint arXiv:2305.09750.*
> >
> > [11] **Wang, K., Babenko, B., & Belongie, S. (2011)*.** End-to-end scene text recognition. Proceedings of the International Conference on Computer Vision (ICCV 2011), Barcelona, Spain.*
> >
> > [12] **Wang, K., & Belongie, S. (2010)*.** Word spotting in the wild. Proceedings of the European Conference on Computer Vision (ECCV 2010), Heraklion, Crete, Greece.*
> >
> > [13] **Sun, Y., Liu, J., Liu, W., Han, J., Ding, E., & Liu, J. (2019).** Chinese street view text: Large-scale Chinese text reading with partially supervised learning. *Proceedings of the 2019 IEEE/CVF International Conference on Computer Vision (ICCV)*, 9085–9094. IEEE.

---

> > > ### Author Response · Authors · 2025-03-27
> > > **Failure Cases discussions**
> > >
> > > We acknowledge the reviewer's suggestion to expand the analysis of failure cases and provide deeper insights. We agree that understanding the model's limitations is important. To address this, **we have significantly expanded Appendix J (Expansion on Failure Cases) based on our observations results reported in the paper**.
> > >
> > > The revised discussion now categorizes common failure modes and provides potential explanations for why they occur:
> > >
> > > 1. Extreme Stroke Width/Style Variations: Discusses challenges with styles significantly outside the training distribution.
> > > 2. Disproportionate Component Scaling: Notes potential negative impacts on semantic understanding if vision/text components are not scaled proportionally.
> > > 3. Vision Head Training Trade-offs: Summarizes the trade-off between detail capture and stability when unfreezing ViT heads.
> > > 4. Untrained Languages: Explains reduced accuracy and limitations of text-conditioned modes when encountering languages absent from training data.
> > > 5. Task Confusion during Decoding: Describes occasional instances where the model confuses tasks (e.g., outputting empty ink for Recognize and Derender) and mentions our fallback mitigation strategy used during inference.

---

> > > > ### Author Response · Authors · 2025-03-27
> > > > **Expansion on Impact Statement**
> > > >
> > > > We thank the reviewer for highlighting the critical importance of discussing ethical implications and broader impact. We agree that this is essential for work involving handwritten text processing and appreciate the detailed suggestions provided.
> > > >
> > > > Thus, **we have updated Appendix N accordingly**.

---

> > > ### Comment · Reviewer_2tDr · 2025-03-31
> > > **Response to Offline vs Online handwriting improvement**
> > >
> > > Thanks for educating me on offline vs online handwriting!
> > > I am still uncertain why converting offline handwritten text to online using the proposed approach cannot serve as a proxy or provide pseudo labels for offline handwriting recognition. Consider a scenario where offline handwritten annotations are unavailable or limited—such cases are common, especially in low-resource languages, but can also be studied for English under this assumption. Any insight or experiments toward this might strengthen the work.

---

> ### Author Response · Authors · 2025-04-01
>
> We sincerely thank the reviewer for their careful follow-up and for taking the time to articulate their remaining uncertainty. We appreciate the opportunity to further clarify the role of InkSight in low-resource settings, particularly in relation to the use of pseudo-labels.
>
> We would be happy to incorporate this discussion into the final version of the paper if the reviewer believes it would contribute meaningfully.
>
> ### Using InkSight for Text Pseudo-Labels
>
> Directly using InkSight's recognized text output (from Recognize and Derender mode) as proxy/pseudo-labels for unlabeled offline images faces a key challenge in truly low-resource scenarios: InkSight's ability to recognize text accurately depends on the priors learned during its multi-task training. If the target low-resource domain (language/script/style) is significantly different from this training data, InkSight's recognition accuracy for that specific domain will likely be low. Using inaccurate text pseudo-labels could harm, rather than help, the training of an offline HWR model. Essentially, for InkSight to provide good text pseudo-labels, it likely needs some relevant signal or prior knowledge about that domain already.
>
> ### Using InkSight's *Rendering/Augmentation*
>
> The most reliable way InkSight currently helps supervised offline HWR in low-resource settings is by leveraging its rendering and data augmentation pipeline (Section 3.2, Appendix F). By applying diverse visual augmentations to the existing scarce labeled offline images, we can significantly expand the visual diversity of the training set.
>
> ### Using InkSight's *Generated Ink* as Pseudo-Label
>
> The reviewer's persistence prompts us to consider another way InkSight's output could serve as a valuable **pseudo-label**, focusing not on the recognized text, but on the **geometric and dynamic properties of the generated digital ink itself**.
>
> Imagine a low-resource offline dataset. Our goal might be unsupervised analysis, like clustering images by semantics or grouping similar-looking words, rather than direct text recognition.
>
> 1. **InkSight's Role:** We can use InkSight (even in Vanilla Derender mode) to convert each image into a digital ink sequence. Even if the model struggles semantically (perhaps applying priors learned from other trained languages), it still attempts to reconstruct the strokes based on visual geometry.
> 2. **Ink as Pseudo-Label:** This generated ink sequence becomes a pseudo-label capturing geometric structure (shape, curvature, connections, dynamics).
> 3. **Clustering/Similarity:** We can then cluster the original images based on the similarity of their corresponding ink pseudo-labels using e.g. sequence comparison methods (e.g., DTW, embeddings).
>
> **One Example in Our Paper:**
>
> Consider the **Korean handwriting example for Small-p** (Fig. 40 compared to Fig. 38, 39). Since Small-p was not trained on Korean, its semantic understanding is limited, and it might apply priors learned from other scripts (like Chinese or English). Despite potential semantic errors, InkSight still attempts to reconstruct some of the strokes correctly. The resulting **ink pseudo-label for this Korean sample reflects geometric structure**, rather than accurate linguistic representation. This geometric structure could still be useful for comparing this sample to other similarly shaped Korean handwritten samples in an unsupervised manner.
>
> **Advantage over OCR:**
>
> One might argue that an OCR model can also faithfully serve as a pseudo label generator, while clustering could be attempted with OCR output, OCR is language-dependent and produces text strings. InkSight's generated **ink is fundamentally stroke-based and language-agnostic.** This allows us to leverage purely **geometric similarities** between handwritten patterns, even across different words or when text recognition fails. This approach doesn't require pairing with any online recognizer and works directly with the inferred stroke structure.
>
> **Impact and Future Work:**
>
> While this doesn't directly train a supervised offline recognizer, analyzing geometric similarities via ink pseudo-labels could inform unsupervised pre-training or data selection for low-resource HWR. Evaluating this novel application (clustering based on ink pseudo-labels) requires new experiments and metrics, placing it beyond the scope of current revisions, but it is an exciting direction inspired by the reviewer and the discussion.

---

> ### Author Response · Authors · 2025-04-03
> **Generalization**
>
> We thank the reviewer for the suggestion to add a more in-depth analysis of the generalization capabilities of our model. We have updated the paper accordingly, and **added Section K in the appendix to discuss the generalization capabilities in more detail.**

---

### Author Response · Authors · 2025-04-30
**Revision Completed with Requested Changes**

Dear Reviewers,

Thank you again for your thoughtful and constructive feedback.

We have uploaded the revised version of our paper, with all changes marked in red. Detailed responses to each of your comments are provided in the corresponding threads.

We appreciate your time and look forward to your continued feedback.

Best regards,
Authors of paper 3944

---

### Author Response · Authors · 2025-06-16

We would like to sincerely thank the reviewers and the Area Chair for their time, thoughtful feedback, and constructive suggestions throughout the review process. We are grateful for the positive assessment of our claims and empirical validation, as well as the recognition of the paper’s potential interest to the handwriting analysis community.

We have carefully revised the manuscript to incorporate all clarifications and improvements discussed during the rebuttal. The updated version of the paper has been uploaded accordingly.

Best regards,

Authors of Paper 3944

---

### Decision · Action_Editor_i3NE · 2025-06-12

**Recommendation:** Accept with minor revision

**Comment:**

The authors are recommended to include all the clarifications mentioned during rebuttal and prepare the final paper taking all the points into account.

**Audience:**

This paper would be of interest to a niche community interested in handwriting analysis and this is a new problem setting that could be of interest to this community.

**Claims And Evidence:**

The claims made in this paper have been thoroughly validated. All reviewers agree with the main claims made in the paper and the empirical evidence provided. The remaining concern of the reviewer who is not convinced is in terms of the specific application for this work. Particularly, the reviewer is not convinced of the utility of the proposed work. However, the response from the authors on this count does address this concern to some extent.